# Do Not Let Low-Probability Tokens Over-Dominate in RL for LLMs

**Zhihe Yang**[1]  **Xufang Luo**[2*]  **Zilong Wang**[2]  **Dongqi Han**[2]  **Zhiyuan He**[2]
**Dongsheng Li**[2]  **Yunjian Xu**[1,3*]
[1]The Chinese University of Hong Kong, Hong Kong SAR, China.
[2]Microsoft Research Asia, Shanghai, China.
[3]The Chinese University of Hong Kong, Shenzhen Research Institute (SZRI), Guangdong, China.
`{zhyang, yjxu}@mae.cuhk.edu.hk`
`{xufluo, wangzilong, dongqihan, dongsli}@microsoft.com`

## Abstract

Reinforcement learning (RL) has become a cornerstone for enhancing the reasoning capabilities of large language models (LLMs), with recent innovations such as Group Relative Policy Optimization (GRPO) demonstrating exceptional effectiveness. In this study, we identify a critical yet underexplored issue in RL training: low-probability tokens disproportionately influence model updates due to their large gradient magnitudes. This dominance hinders the effective learning of high-probability tokens, whose gradients are essential for LLMs' performance but are substantially suppressed. To mitigate this interference, we propose two novel methods: *Advantage Reweighting* and *Low-Probability Token Isolation (Lopti)*, both of which effectively attenuate gradients from low-probability tokens while emphasizing parameter updates driven by high-probability tokens. Our approaches promote balanced updates across tokens with varying probabilities, thereby enhancing the efficiency of RL training. Experimental results demonstrate that they substantially improve the performance of GRPO-trained LLMs, achieving up to a 46.2% improvement in K&K Logic Puzzle reasoning tasks. Our implementation is available at `https://github.com/zhyang2226/AR-Lopti`.

## 1 Introduction

The reasoning capabilities of large language models (LLMs) have recently achieved a milestone breakthrough with the integration of reinforcement learning (RL) during post-training phase (Jaech et al., 2024; Guo et al., 2025; Team et al., 2025). Intuitively, the vast vocabulary size and the auto-regressive generation mechanism of LLMs pose significant challenges for effective exploration due to the exponentially large state space. DeepSeek-R1 (Guo et al., 2025) eliminates this bias, demonstrating that 'simple RL with rule-based reward' can significantly enhance the reasoning abilities of LLMs without relying on scaffolding techniques such as Monte Carlo Tree Search (MCTS) (Xie et al., 2024b; Chen et al., 2024) or Progress Reward Modeling (PRM) (Lightman et al., 2024; Wang et al., 2024). Moreover, they introduce a novel algorithm, Group Relative Policy Optimization (GRPO) (Shao et al., 2024), which has proven highly effective in the domains of mathematics and code, inspiring numerous follow-up studies.

Yu et al. (2025) and Liu et al. (2025) consistently report that GRPO training leads to progressively longer response lengths, while the increase does not correspond to a proportional improvement in the model's performance. They attribute this trend to the bias in update weights related to response length inherent in GRPO's objective. Xiong et al. (2025) conduct comparison between GRPO and Proximal Policy Optimization (PPO). They find that the instability of PPO, compared to GRPO, arises from its unnecessary bias toward entirely incorrect responses on overly difficult prompts. In contrast, GRPO mitigates this issue by discarding such prompts through a within-prompt normalization operation. These findings highlight the substantial impact of update bias on training outcomes.

---

[*]Corresponding authors.

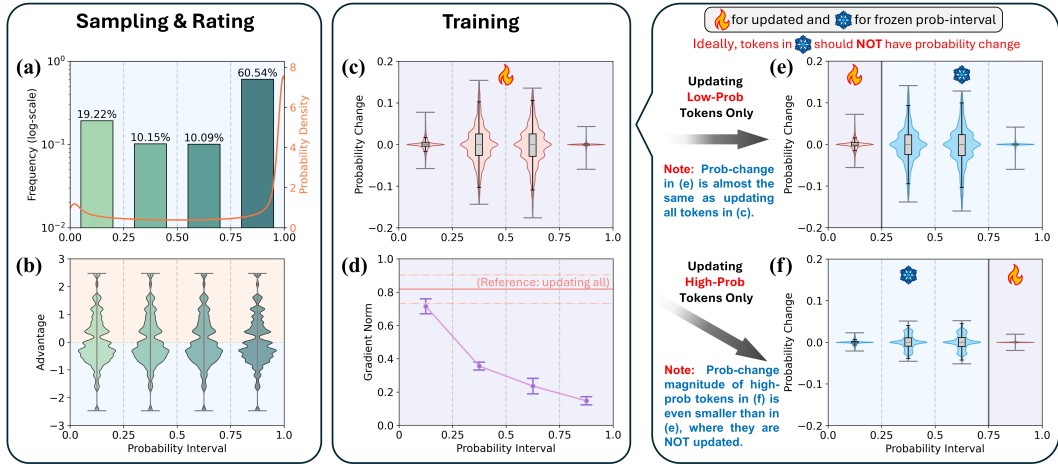

Figure 1: Experimental analysis on the K&K Logic Puzzle dataset during GRPO training of `Qwen2.5-7B-Instruct-1M`. Tokens are divided into four groups based on probability quartiles. (a) Token probability distribution and (b) corresponding advantages. (c) Token probability changes after updates (using SGD with lr=1e-3) and (d) gradient norms for each probability group. Effects of selective updates: (e) Probability changes when only tokens in the lowest quartile (probability < 0.25) are updated, and (f) when only tokens in the highest quartile (probability > 0.75) are updated. To ensure clarity, the top 1% of outlier samples in the violin plots for token probability changes are excluded. Results are averaged over 10 randomly sampled batches.

In this study, we identify another important source of update bias in RL training, which is orthogonal to aforementioned ones and has rarely been noted in prior research. This bias arises from the gradient perspective and is strongly correlated with the token probabilities. As shown in Figure 1, during GRPO training, tokens are divided into four groups based on probability quartiles. The policy gradient is conducted with the advantage presented in Figure 1(b). Figure 1(d) shows that low-probability tokens generate disproportionately larger gradients compared to high-probability ones. Since each RL update involves hundreds of thousands of tokens with interacting gradients, low-probability tokens are expected to have a greater influence. To verify this, we independently update tokens from the lowest and highest quartiles, as shown in Figures 1(e) and (f). The pattern in (e) closely matches (c), while (f) looks significantly different. Interestingly, in (e), even though high-probability tokens were not updated, their probabilities changed more significantly than when they were updated (as shown in (f)). Thus, we conclude that **low-probability tokens dominate model updates** during RL training and that **this dominance may impede the precise adjustment of the probability distribution across all tokens**. Notably, we observe that high-probability tokens are much less likely to be updated in the correct direction compared to low-probability tokens (cf. Figure 3).

By deriving the gradients induced by individual tokens, we reveal a key property of RL training that explains the phenomenon illustrated in Figure 1. Specifically, for an LLM comprising a benign neural network, the gradient norm of any intermediate activation corresponding to a single token is bounded between two values proportional to $(1 - \pi)$, where $\pi$ is the token's probability. This property underscores that tokens with lower probabilities result in larger gradient magnitudes, whereas tokens with probabilities approaching 1 yield gradients that are nearly negligible.

To mitigate the over-dominance of low-probability tokens and promote more efficient updates, we propose two simple yet effective methods: *Advantage Reweighting*, which reduces the weight assigned to low-probability tokens, and *Low-Probability Token Isolation (Lopti)*, which separates low-probability tokens and updates them prior to high-probability tokens. Both methods attenuate gradients from low-probability tokens while emphasizing parameter updates driven by high-probability tokens. Notably, the first one incurs almost no additional computational cost. These methods can be applied independently, each providing benefits, or together, with the potential for further performance improvements. Experimental results demonstrate the effectiveness of the proposed methods across various datasets. In particular, on K&K Logic Puzzle dataset, they enhance the performance of naive GRPO (trained from `Qwen2.5-3B-Instruct`) by 35.9% and 38.5%, respectively, and by 46.2% when used together.

In summary, our contributions are threefold: (1) We identify a critical issue in RL training for LLMs that has received limited attention: low-probability tokens disproportionately dominate the updates due to their large gradient contributions. (2) We provide a concise theoretical explanation for this phenomenon. (3) Based on the identified issue, we propose two simple yet effective methods, which significantly improve the downstream performance of GRPO-trained LLMs across various datasets.

## 2 RELATED WORK

As a fundamental technique driving recent advancements in LLMs, reinforcement learning is attracting increasing attention from researchers. In this section, we provide a concise overview on the development of RL in the context of LLMs.

RL was pioneered by OpenAI as the final step of post-training to further align fine-tuned large models with human preferences (Christiano et al., 2017; Ziegler et al., 2019; Stiennon et al., 2020; Ouyang et al., 2022). By leveraging vast amounts of human preference data and stable RL algorithms such as PPO (Schulman et al., 2017), numerous enterprise-level language models have benefited from this approach and have been widely adopted. Notable examples include ChatGPT (Brown et al., 2020; Achiam et al., 2023), LLaMA (Touvron et al., 2023a;b; Dubey et al., 2024), Qwen (Bai et al., 2023; Chu et al., 2023; Yang et al., 2024), Gemini (Team et al., 2023; 2024), and Claude (Anthropic, 2024). Nevertheless, the challenges of collecting high-quality data that accurately reflect human preferences, the limited performance of open-source LLMs, and the computationally intensive training requirements of PPO-like online RL algorithms pose significant barriers for further exploring RL's potentiality in the domain of LLMs. Most studies have focused on simplifying RL algorithms and directly leveraging preference data to optimize models. Representative works include Direct Preference Optimization (DPO) (Rafailov et al., 2023; 2024), related analyses (Xu et al., 2024b; Zhong et al., 2024; Ren & Sutherland, 2025), and improved variants such as ORPO (Hong et al., 2024), CPO (Xu et al., 2024a) and SimPO (Meng et al., 2024).

Recently, the emergence of long-chain-of-thought (CoT) (Wei et al., 2022) reasoning and its integration into both pre-training and post-training processes have significantly advanced the foundational capabilities of LLMs. OpenAI-o1 (Jaech et al., 2024) was the first to demonstrate the remarkable potential of combining RL with CoT, enabling LLMs to surpass human cognitive abilities and tackle complex mathematical and coding tasks for the first time. Shortly thereafter, Deepseek-R1 (Guo et al., 2025) fully harnessed the potential of RL+CoT through a simple yet highly effective reinforcement learning algorithm GRPO (Shao et al., 2024). Their findings revealed that LLMs exhibit human-like 'aha moments' during RL training. This achievement quickly garnered significant attention, inspiring extensive replication efforts (Luo et al., 2025; Xie et al., 2025; Hu et al., 2025; Zeng et al., 2025) stimulating further research on enhancing GRPO (Yu et al., 2025; Liu et al., 2025) and PPO (Yuan et al., 2025; Shi et al., 2025), as well as comparative analyses between the two (Xiong et al., 2025). Nevertheless, most existing improvement solutions focus on enhancing sample quality, balancing response length, and preventing entropy collapse. To the best our knowledge, this work is the first to improve RL training from the gradient-disproportionality perspective.

## 3 PRELIMINARY

**Large Language Models.** Most existing LLMs are based on a transformer decoder-only architecture (Vaswani et al., 2017), typically denoted as $\pi_\theta$, where $\theta \in \mathbb{R}^d$ represents the model parameters. The fundamental unit of LLMs is the token, a discrete textual element that may correspond to a word, subword, or character, and is drawn from a finite vocabulary $\mathcal{V} = \{v^1, \ldots, v^N\}$, where $N$ denotes the vocabulary size. During text generation, the model outputs a probability distribution over the vocabulary, conditioned on the given prompt $\boldsymbol{q}$ and the sequence of previously generated tokens $\boldsymbol{o}_{<t}$. The next token $o_t$ is then sampled from this distribution, expressed mathematically as $o_t \sim \pi_\theta(\cdot|\boldsymbol{q}, \boldsymbol{o}_{<t})$. The generation process is autoregressive, proceeding iteratively until either an end-of-sentence (EOS) token is produced or a predefined maximum sequence length $t_{max}$ is reached. The resulting sequence of tokens is denoted as $\boldsymbol{o}$.

Practical LLMs are often required to align with human preferences or exhibit strong reasoning capabilities, which cannot be easily achieved through naive pre-training and supervised fine-tuning. If a reward function $r(\boldsymbol{q}, \boldsymbol{o})$ is available to quantitatively capture these objectives, the optimization of

an LLM can be formulated as a reinforcement learning task. In this framework, the generation of each token is treated as an action, while the prompt and the previously generated tokens are treated as the state. Accordingly, the optimization objective of the LLM is expressed as $\max_\theta \mathbb{E}_{\boldsymbol{q}\sim\mathcal{D}, \boldsymbol{o}\sim\pi_\theta}[r(\boldsymbol{q}, \boldsymbol{o})]$, where $\mathcal{D}$ is pre-collected dataset.

**Group Relative Policy Optimization.** As a widely used algorithm in early-stage research, PPO (Schulman et al., 2017) requires a value model with as many—or even more—parameters as the model being trained. The value model must be trained in conjunction with LLMs, and its initialization adds complexity and uncertainties to the RL training process. To address these challenges, DeepSeek introduces GRPO (Liu et al., 2025), which eliminates the need for a value model entirely by estimating value through group-relative comparison. Specifically, for each question $\boldsymbol{q}$, GRPO samples a group of outputs $\{\boldsymbol{o}_1, \boldsymbol{o}_2, \ldots, \boldsymbol{o}_G\}$ and estimate the expected return under the question through $V(\boldsymbol{q}) = \text{mean}(r(\boldsymbol{q}, \boldsymbol{o}_1), r(\boldsymbol{q}, \boldsymbol{o}_2), \ldots)$. During the training process, the estimated advantage is set to be consistence within each responses ($\hat{A}_{i,t} = \hat{A}_i$), and is calculated through $\hat{A}_i = \frac{r(\boldsymbol{q}, \boldsymbol{o}_i) - V(\boldsymbol{q})}{\text{std}(r(\boldsymbol{q}, \boldsymbol{o}_1), r(\boldsymbol{q}, \boldsymbol{o}_2), \ldots)}$. Compared to PPO, GRPO reduces GPU memory overhead by 50% and decreases single-step RL training time by over 60% (Xie et al., 2025). In this work, we adopt a variant of GRPO to optimize the policy model $\pi_\theta$. The optimization objective is expressed as follows:

$$J_{GRPO}(\theta) = \mathbb{E}_{\boldsymbol{q}\sim\mathcal{D}, \{\boldsymbol{o}_i\}_{i=1}^G \sim \pi_{old}}$$

$$\frac{1}{\sum_{i=1}^G |\boldsymbol{o}_i|} \sum_{i=1}^G \sum_{t=1}^{|\boldsymbol{o}_i|} \left\{ \min\left[ r_{i,t}(\theta)\hat{A}_{i,t}, \text{clip}(r_{i,t}(\theta); 1-\epsilon_l, 1+\epsilon_h)\hat{A}_{i,t} \right] - \beta \, \mathbb{D}_{\text{KL}}\left[\pi_\theta \| \pi_{ref}\right] \right\}$$

$$\text{with } r_{i,t}(\theta) = \frac{\pi_\theta(o_{i,t}|\boldsymbol{q}, \boldsymbol{o}_{i,<t})}{\pi_{old}(o_{i,t}|\boldsymbol{q}, \boldsymbol{o}_{i,<t})}, \text{and } \mathbb{D}_{\text{KL}}\left[\pi_\theta \| \pi_{ref}\right] = \frac{\pi_{ref}(o_{i,t}|\boldsymbol{q}, \boldsymbol{o}_{i,<t})}{\pi_\theta(o_{i,t}|\boldsymbol{q}, \boldsymbol{o}_{i,<t})} - \log\frac{\pi_{ref}(o_{i,t}|\boldsymbol{q}, \boldsymbol{o}_{i,<t})}{\pi_\theta(o_{i,t}|\boldsymbol{q}, \boldsymbol{o}_{i,<t})} - 1,$$

$$(1)$$

where $\pi_{old}$ denotes the policy used to sample the responses, $\pi_{ref}$ represents the initial policy prior to RL training, and $\epsilon_l, \epsilon_h, \beta$ are manually defined hyperparameters. Note that the original implementation of GRPO normalizes the token update weights based on the response length, which introduces a significant bias toward shorter responses during updates. In line with verl (Sheng et al., 2025) and most follow-up work (Zeng et al., 2025; Liu et al., 2025), we remove this operation and conduct normalization among all tokens within the same query-batch.

## 4 METHODOLOGY

### 4.1 EXPLANATION ON LOW-PROBABILITY TOKENS' DOMINANCE

In this section, we provide a theoretical explanation for why tokens with lower probabilities tend to dominate updates during RL training. The learning objective in equation 1 can be interpreted as a weighted cross-entropy loss. For simplicity, we use the notation $\pi(o_{i,t})$ to denote $\pi(o_{i,t}|\boldsymbol{q}, \boldsymbol{o}_{i,<t})$. By evaluating the gradient, we obtain the following expression (cf. Appendix A.1 for derivation):

$$\nabla_\theta J_{GRPO}(\theta) = \mathbb{E}_{\boldsymbol{q}\sim\mathcal{D}, \{\boldsymbol{o}_i\}_{i=1}^G \sim \pi_{old}} \frac{1}{\sum_{i=1}^G |\boldsymbol{o}_i|} \sum_{i=1}^G \sum_{t=1}^{|\boldsymbol{o}_i|}$$

$$\underbrace{\left[ \frac{\pi_\theta(o_{i,t})}{\pi_{old}(o_{i,t})} \hat{A}_{i,t} \cdot \mathbb{I}_{\text{trust}}\left(\frac{\pi_\theta(o_{i,t})}{\pi_{old}(o_{i,t})}, \hat{A}_{i,t}\right) + \beta \frac{\pi_{ref}(o_{i,t})}{\pi_\theta(o_{i,t})} - \beta \right]}_{w_{i,t}} \cdot \nabla_\theta \log \pi_\theta(o_{i,t}),$$

$$(2)$$

$$\text{where } \mathbb{I}_{\text{trust}}\left(\frac{\pi_\theta(o_{i,t})}{\pi_{old}(o_{i,t})}, \hat{A}_{i,t}\right) = \begin{cases} 0 & \begin{cases} \text{if } \hat{A}_{i,t} > 0 \text{ and } \frac{\pi_\theta(o_{i,t})}{\pi_{old}(o_{i,t})} > 1 + \epsilon_h \\ \text{if } \hat{A}_{i,t} < 0 \text{ and } \frac{\pi_\theta(o_{i,t})}{\pi_{old}(o_{i,t})} < 1 - \epsilon_l \end{cases} \\ 1 & \text{otherwise} \end{cases}.$$

We represent LLM as a composite function $f = f_L \circ f_{L-1} \circ \cdots \circ f_1$, where each $f_\ell$ (with $\ell \in \{1, \ldots, L\}$) corresponds to a distinct layer of the network. Let $a_{\ell-1}$ denote the input and $a_\ell$ denotes the output of $\ell$th layer. We further define the Jacobian matrix of the $\ell$th layer with respect to its input as $J_\ell := \frac{\partial f_\ell(a_{\ell-1})}{\partial a_{\ell-1}}$.

**Assumption 4.1.** *For every layer, the Jacobian $J_\ell$ is well-defined and the $f_\ell$ is locally differentiable. Furthermore, assume that for each layer, there exist two constants $c_\ell > 0$ and $d_\ell > 0$ such that*

$\sigma_{\min}(J_\ell) \geq c_\ell$ and $\sigma_{\max}(J_\ell) \leq d_\ell$, where $\sigma_{\min}(\cdot)$ and $\sigma_{\max}(\cdot)$ denote the minimum and maximum singular values of the given matrix, respectively.

Assumption 4.1 is not restrictive, as it aligns with the standard design and training principles of neural-networks, ensuring stable gradients flow through well-defined and non-degenerate Jacobians.

**Proposition 4.2.** *Under Assumption 4.1, let $\delta_\ell(o_{i,t}) := \nabla_{a_\ell} J_{GRPO}(o_{i,t})$ denote the gradient of the GRPO objective with respect to activation $a_\ell$ at any layer for a single token $o_{i,t}$. Let $\|\cdot\|$ denote the spectral norm, and define the vocabulary size as $N$. Then, for each layer $\ell$, the following inequalities always hold:*

$$\prod_{j=\ell+1}^{L} c_j \cdot |w_{i,t}| \cdot \sqrt{\frac{N}{N-1}} \cdot \left(1 - \pi_\theta(o_{i,t})\right) \leq \|\delta_\ell(o_{i,t})\| \leq \prod_{j=\ell+1}^{L} d_j \cdot |w_{i,t}| \cdot \sqrt{2} \cdot \left(1 - \pi_\theta(o_{i,t})\right). \quad (3)$$

Refer to Appendix A.2 for the detailed proof. Proposition 4.2 demonstrate that, for a single token, the gradient norm with respect to activation $a_\ell$ at any layer is bounded. Specifically, it is confined within the truncated conical region illustrated in Figure 2. In equation 3, apart from the term $(1-\pi_\theta(o_{i,t}))$, all other components in these bounds can be regarded as constant. (Although $w_{i,t}$ depends on $\pi_\theta(o_{i,t})$, it is approximately equal to $\hat{A}_{i,t}$ in most cases.) This result highlights that *tokens with lower probabilities lead to larger gradient magnitudes, whereas tokens with probabilities approaching 1 produce gradients that are nearly zero.* The experimental evidence presented in Figure 1 corroborates this relationship, demonstrating a roughly proportional correspondence between the gradient norm of all LLM parameters and $(1 - \pi_\theta(o_{i,t}))$.

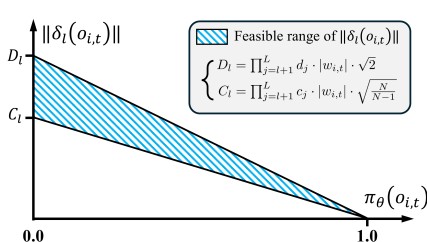

Figure 2: Diagram of Proposition 4.2.

Notably, during the RL training process, the gradients are averaged over hundreds of thousands of tokens for each update. Typically, the gradients are not sparsely distributed, leading to mutual influence among them. In such cases, low-probability tokens tend to dominate the gradient updates. Nevertheless, the gradients of high-probability tokens are equally important and should not be neglected (see Section 5.3 for details). To the best of our knowledge, no prior study has explicitly investigated the gradient interference between low-probability and high-probability tokens.

### 4.2 MITIGATING THE OVER-DOMINANCE OF LOW-PROBABILITY TOKENS

**Adverse Effect of the Dominance.** A natural question arises: what are the consequences if the gradient of low-probability tokens over-dominates the update process? Experimental results in Xiong et al. (2025) suggest that positive samples (i.e., responses/tokens with an advantage greater than 0) play a more significant role than those negative ones. Theoretically, the probability of tokens with positive advantage should increase after each update. Thus, we record the proportion of positive tokens with increased probabilities during a single RL training step, as shown in Figure 3. In line with expectations, as the probability of a token grows, the proportion of updates in the correct direction decreases. In particular, the proportion of correct update directions for tokens with probability greater than 0.75 is even slightly less than 50%. To mitigate the over-dominance of low-probability tokens and promote more efficient updates for high-probability tokens, we introduce the following two methods.

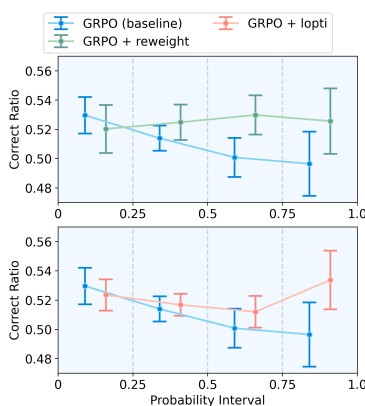

Figure 3: The proportion of positive tokens updated in the correct direction for different updating methods, under the same experimental settings as in Figure 1.

**Advantage Reweighting.** A straightforward approach to address this issue is to reweight the advantage of tokens based on their probabilities. Specifically, we re-calculate the advantage

---

**Algorithm 1** GRPO with Advantage Reweighting and Low-Probability Token Isolation

---

**Require:** Initial LLM $\pi_\theta = \pi_{ref}$, datasets $\mathcal{D} = \{q\}$, reward function $r(q, o)$, reweighting hyperparamter $\alpha$,
isolation threshold $\eta$

1: **for** each dataset epoch **do**
2:     **for** each RL step, sample $\{q\}^M \sim \mathcal{D}$ **do**
3:       Auto-regress sampling $G$ responses $\{o_i\}_{i=1}^G$ for each question within $\{q\}^M$
4:       Record the old probability for each token $\pi_{old}(o_{i,t}) = \pi_\theta(o_{i,t})$
5:       Calculate the reward for each response with reward function $r(q, o_i)$
6:       Calculate the advantage for each token (response) through $\hat{A}_{i,t} = \hat{A}_i = \frac{r(q, o_i) - \text{mean}\{r(q, o_i)\}_{i=1}^G}{\text{std}\{r(q, o_i)\}_{i=1}^G}$
7:       Reweight Advantage through equation 4
8:       ~~**for** each RL epoch, sample mini_batch $\sim \{q, \{\{\hat{A}_{i,t}, \pi_{old}(o_{i,t})\}_{t=1}^{|o_i|}\}_{i=1}^G\}^M$ **do**~~
9:         ~~Update the policy $\pi_\theta$ with mini_batch through equation 1~~
10:      ~~**end for**~~
11:      Record the old Advantage $\hat{A}_{i,t}^{old} = \hat{A}_{i,t}$
12:      Mask high-probability tokens through $\hat{A}_{i,t} = \hat{A}_{i,t}^{old} \odot \mathbb{I}(\pi_{old}(o_{i,t}) \leq \eta)$
13:      **for** each RL epoch, sample mini_batch $\sim \{q, \{\{\hat{A}_{i,t}, \pi_{old}(o_{i,t})\}_{t=1}^{|o_i|}\}_{i=1}^G\}^M$ **do**
14:         Update the policy $\pi_\theta$ with mini batch through equation 1
15:      **end for**
16:      Mask low-probability tokens $\hat{A}_{i,t} = \hat{A}_{i,t}^{old} \odot (1 - \mathbb{I}(\pi_{old}(o_{i,t}) \leq \eta))$
17:      **for** each RL epoch, sample mini_batch $\sim \{q, \{\{\hat{A}_{i,t}, \pi_{old}(o_{i,t})\}_{t=1}^{|o_i|}\}_{i=1}^G\}^M$ **do**
18:         Update the policy $\pi_\theta$ with mini batch through equation 1
19:      **end for**
20:     **end for**
21: **end for**
22: **return** Final policy $\pi_\theta$

---

of each token as follows:

$$\hat{A}_{i,t} = [\alpha \cdot \pi_\theta(o_{i,t}) + (1 - \alpha)] \cdot \hat{A}_{i,t}, \tag{4}$$

where $\alpha \in [0, 1]$ is a manually-defined hyperparameter. This formulation assigns linearly smaller update weights to tokens with lower probabilities. As shown in the upper panel of Figure 3, it can significantly reduce the errors in update directions for positive high-probability tokens.

**Low-Probability Tokens Isolation (Lopti).** In addition to *Advantage Reweighting*, we also explored an alternative method, referred to as *Lopti*. Specifically, for a sampled mini-batch in RL, we predefine a probability threshold $\eta \in (0, 1)$ to divide tokens into two groups: low-probability tokens and high-probability tokens. We first update the low-probability tokens, followed by the high-probability tokens. For detailed implementation, please refer to lines 11–19 of Algorithm 1. With a universal hyperparameter setting of $\eta = 0.5$, this method achieves a comparable effect to *Advantage Reweighting*, as shown in the lower panel of Figure 3.

The intuition behind *Lopti* is as follows: during the first stage, updates on low-probability tokens indirectly influence the distribution of the remaining high-probability tokens that have not yet been updated (as in Figure 1(e)). If a positive high-probability token is affected in the correct direction (i.e., its probability increases), its gradient becomes smaller in the subsequent stage when high-probability tokens are updated. Conversely, if its probability decreases, its gradient will dominate within the high-probability token group, thereby receiving greater attention during the update process. Note that the order of updates cannot be reversed. The corresponding ablation is presented in Section 5.3.

It is worth noting that *Advantage Reweighting* and *Lopti* can operate concurrently and may even lead to further improved downstream performance. In Algorithm 1, we detail how to integrate these two techniques with GRPO. Note that the original GRPO update step (the gray section with strikethrough in lines 8–10) should be skipped if *Lopti* is activated. The computational cost requirements are detailed in Appendix C.2. Since *Lopti* splits the tokens and performs updates twice, it results in higher computational costs, which is a limitation of our method (cf. Appendix F).

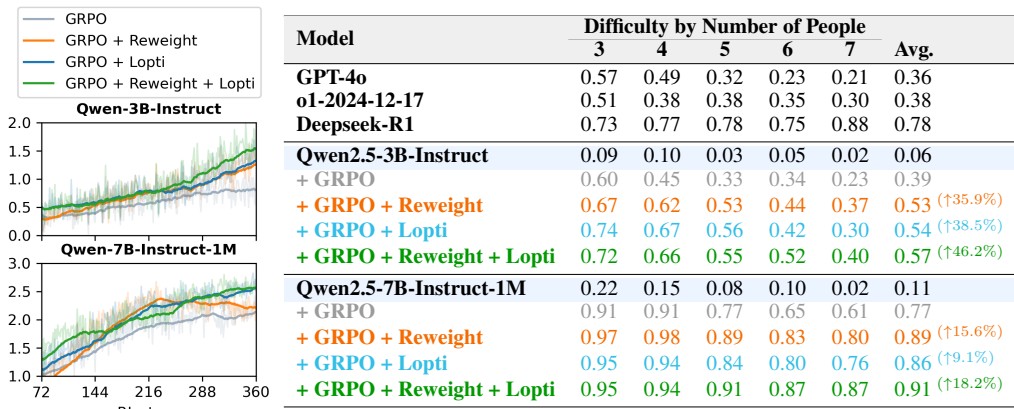

| Model | Difficulty by Number of People | | | | | |
|---|---|---|---|---|---|---|
| | 3 | 4 | 5 | 6 | 7 | Avg. |
| **GPT-4o** | 0.57 | 0.49 | 0.32 | 0.23 | 0.21 | 0.36 |
| **o1-2024-12-17** | 0.51 | 0.38 | 0.38 | 0.35 | 0.30 | 0.38 |
| **Deepseek-R1** | 0.73 | 0.77 | 0.78 | 0.75 | 0.88 | 0.78 |
| **Qwen2.5-3B-Instruct** | 0.09 | 0.10 | 0.03 | 0.05 | 0.02 | 0.06 |
| + GRPO | 0.60 | 0.45 | 0.33 | 0.34 | 0.23 | 0.39 |
| **+ GRPO + Reweight** | 0.67 | 0.62 | 0.53 | 0.44 | 0.37 | 0.53 (↑35.9%) |
| **+ GRPO + Lopti** | 0.74 | 0.67 | 0.56 | 0.42 | 0.30 | 0.54 (↑38.5%) |
| **+ GRPO + Reweight + Lopti** | 0.72 | 0.66 | 0.55 | 0.52 | 0.40 | 0.57 (↑46.2%) |
| **Qwen2.5-7B-Instruct-1M** | 0.22 | 0.15 | 0.08 | 0.10 | 0.02 | 0.11 |
| + GRPO | 0.91 | 0.91 | 0.77 | 0.65 | 0.61 | 0.77 |
| **+ GRPO + Reweight** | 0.97 | 0.98 | 0.89 | 0.83 | 0.80 | 0.89 (↑15.6%) |
| **+ GRPO + Lopti** | 0.95 | 0.94 | 0.84 | 0.80 | 0.76 | 0.86 (↑9.1%) |
| **+ GRPO + Reweight + Lopti** | 0.95 | 0.94 | 0.91 | 0.87 | 0.87 | 0.91 (↑18.2%) |

Figure 4: Experimental results on the K&K Logic Puzzles benchmark. For *Advantage Reweight*, $\alpha = 0.3$, and for *Lopti*, $\eta = 0.5$. The reward curve during training (left) is truncated to exclude the first epoch and smoothed with an exponential moving average (coefficient: 0.95). The evaluation accuracy on the test set (right) are averaged over the last three checkpoints to mitigate randomness.

## 5 EXPERIMENTAL RESULTS

To validate the effectiveness of our proposed method, we first conduct experiments on the Knights and Knaves (K&K) Logic Puzzles dataset (Xie et al., 2025; 2024a) using GRPO, as described in Section 5.1. We then extend the experiments to the math-related dataset (Luo et al., 2025; Shi et al., 2025), as detailed in Section 5.2. Finally, we present a series of critical ablation studies, as outlined in Section 5.3. Note that our methods are not restricted to GRPO and hold great potential across all Policy-Gradient based RL algorithms. For experiments utilizing REINFORCE++ (Hu, 2025), please refer to Appendix D.

### 5.1 EXPERIMENTS ON K&K LOGIC PUZZLES

The K&K logic puzzles, first aggregated into a benchmark for LLMs by Xie et al. (2024a), are a class of reasoning problems rooted in classical logic game (Smullyan, 1986; Johnson-Laird & Byrne, 1990). These puzzles involve a fictional scenario where inhabitants of an island are either Knights, who always tell the truth, or Knaves, who always lie. The objective is to determine the identity of each inhabitant (Knight or Knave) based on a set of statements they make about themselves and others. Please refer to Appendix C.1.1 for detailed introduction. The K&K logic puzzles are highly challenging, with only the most advanced LLMs demonstrating strong performance (Xie et al., 2024a). Additionally, it is not exposed in the model's pre-training phase, allowing the model to demonstrate continual learning behavior during training. As training progresses, both the training reward and test accuracy gradually improve, rather than converging rapidly. These characteristics make this benchmark an ideal choice for verifying RL performance.

Following Logic-RL (Xie et al., 2025), we construct the training set by combining logic puzzles with 3 to 7 players and adopt its rule-based reward function, which consists of two components: (1) Format score, assigned 1 if the model provides CoT reasoning within `<think></think>` tags and the final answer within `<answer></answer>` tags, and -1 otherwise; (2) Answer reward, assigned 2 for a perfect match with the ground truth, -1.5 for partial correctness, and -2 for an completely incorrect answer. We use `Qwen2.5-3B-Instruct` and `Qwen2.5-7B-Instruct-1M` as starting points. Without employing curriculum learning, we directly expose the model to the mixed training set and train it for a total of 5 epochs. The experimental results are reported in Figure 4. Detailed hyperparameter settings are provided in Appendix B, and comprehensive experimental records can be found in Appendix C.1.1.

During the early stages of GRPO training, the reward increases rapidly, but the growth slows significantly after the first epoch. Subsequently, the improvements introduced by *Advantage Reweighting* and *Lopti* become progressively more evident, particularly after 4 epochs. Interestingly, for simpler tasks (involving fewer players), the performance gap between the baseline GRPO and the GRPO enhanced with *Advantage Reweighting* and/or *Lopti* is minimal. However, for more complex tasks

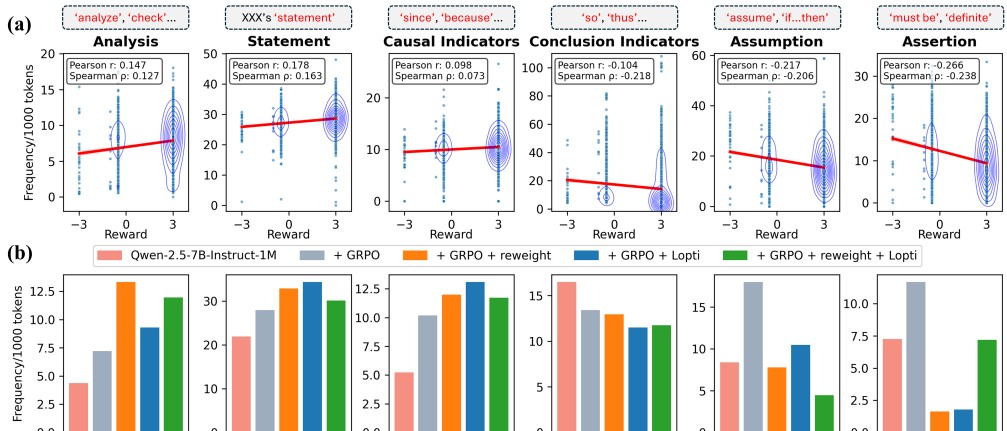

Figure 5: (a) The relationship between the frequency of six categories of inference-related words and the corresponding sample rewards for Qwen-2.5-7B-Instruct-1M trained with naive GRPO. The Pearson correlation coefficient ($r$) and Spearman rank correlation coefficient ($\rho$) are annotated. (b) A comparison of the frequency of the six categories of words across the starting point (Qwen-2.5-7B-Instruct-1M), naive GRPO, and GRPO enhanced with Advantage Reweighting and/or Lopti.

with more players, the performance gap becomes significant. In challenging tasks, positive samples are typically fewer and thus more valuable. As analyzed in Section 4.2, high-probability tokens in these rare positive samples are not effectively amplified under standard GRPO training. Our method addresses this limitation, thereby resulting in substantial performance improvements.

In addition, we perform a linguistic analysis to investigate the correlation between the model's reasoning behavior and its final performance. Specifically, we use the model trained with naive GRPO to generate responses for the 500 prompts in the test set, sampling 8 responses per prompt, resulting in a total of 4,000 samples. For these samples, we analyze the frequency of six categories of inference-related words (see Appendix C.1.1 for details) and their corresponding rule-based rewards, as illustrated in Figure 5(a). The analysis reveals a positive correlation between the frequency of words in the categories *Analysis*, *Statement*, and *Causal Indicators* and the samples' rewards. Conversely, the frequency of words in the categories *Conclusion Indicator*, *Assumption*, and *Assertion* exhibits a negative correlation with the rewards.

It is worth noting that the statistical patterns observed in these six categories of words indirectly highlight the enhancement effects of our proposed *Advantage Reweighting* and/or *Lopti* mechanisms on GRPO training, as shown in Figure 5(b). Notably, the frequency of words positively correlated with reward in the samples generated by our method is significantly higher than that of the baseline, while the frequency of words negatively correlated with reward is substantially lower.

## 5.2 EXPERIMENTS ON MATH-RELATED DATASETS

To assess the generalization capability of our proposed methods, we conduct additional experiments on math-related datasets. Consistent with the majority of prior studies, we utilize `Qwen2.5-7B` as the base model and employ a straightforward rule-based reward. Specifically, a score of 1 is assigned for completely correct answers, while a score of 0 is given for all other cases. We experiment with two different datasets. The first one is a subset containing 10k problems introduced by AdaRFT (Shi et al., 2025), which is sampled from DeepScaleR (Luo et al., 2025). This dataset, referred to as DSR-Uniform, evenly covers problems across all difficulty levels and is specifically designed for `Qwen2.5-7B`. We train this dataset for 5 epochs. The second one is a dataset containing 57k problems introduced by Open-Reasoner-Zero (ORZ) (Liu et al., 2025). For this dataset (ORZ), we train for 1 epoch. Apart from the number of training epochs, all other hyperparameters (cf. Appendix B) are kept consistent across both datasets.

We evaluate the LLMs after training on five benchmarks: Olympiad Bench (He et al., 2024), Minerva (Lewkowycz et al., 2022), MATH-500 (Hendrycks et al., 2021), AMC 2022-2023, and AIME 2024. For the first three benchmarks, we use greedy sampling for evaluation. For the last

Table 1: Experimental results on math-related datasets (DSR for DeepScaleR and ORZ for Open-Reasoner-Zero). For *Advantage Reweight*, $\alpha$ is set to 0.1, and for *Lopti*, $\eta$ is set to 0.5. The evaluation accuracy(%) are averaged over the last three checkpoints to mitigate randomness.

| Dataset | Algorithms | Olympiad Bench | Minerva | MATH 500 | AMC avg@16 | AIME24 pass@16 | AIME24 avg@16 | Avg. all |
|---|---|---|---|---|---|---|---|---|
| **Qwen2.5-7B** | | 27.64 | 18.38 | 63.00 | 22.21 | 30.00 | 5.00 | 27.71 |
| **DSR Uniform** | + GRPO | 36.50 | 29.66 | 74.67 | 47.72 | 28.89 | 16.46 | 38.98 |
| | + GRPO + Reweight | 37.00 | 29.66 | 75.47 | 48.32 | 35.56 | 14.03 | 40.01 |
| | + GRPO + Lopti | 36.60 | 30.27 | 76.53 | 47.69 | 32.22 | 14.24 | 39.59 |
| **ORZ** | + GRPO | 38.23 | 27.69 | 78.33 | 49.57 | 32.22 | 12.92 | 39.83 |
| | + GRPO + Reweight | 40.81 | 29.04 | 77.80 | 49.07 | 33.33 | 16.46 | 41.09 |
| | + GRPO + Lopti | 38.63 | 29.78 | 78.53 | 47.29 | 34.44 | 15.28 | 40.66 |

two benchmarks, following prior works, we sample 16 responses for each question and report the average accuracy (avg@16). Notably, AIME 2024 is extremely challenging; therefore, we also report pass@16, which considers a question correctly answered if at least one of the 16 responses is correct.

The experimental results are summarized in Table 1. In contrast to the continual learning behavior observed in the K&K Logic Puzzle dataset, the test accuracy curve on the math-related dataset converges to a specific value within 100 steps and subsequently exhibits only minor fluctuations. Despite this, the improvements introduced by our *Advantage Reweighting* and *Lopti* remain observable. It is worth noting that the combined application of these two techniques does not result in further performance gains; therefore, we recommend using them individually for optimal results. The underlying reasons for this phenomenon are discussed in Appendix C.3. For detailed experimental records, please refer to Appendix C.1.2.

## 5.3 ABLATION STUDIES

To better convey our motivation and demonstrate the effectiveness of the proposed methods, we perform ablation studies on the K&K Logic Puzzles dataset. The key conclusions derived from these studies are summarized in the following three points.

- **High-probability tokens matter in RL training.** Although the results in Figure 1 and Figure 3 suggest that the gradients of high-probability tokens are almost suppressed by low-probability tokens during updates, the high-probability tokens remain crucial and cannot be disregarded. As shown in Figure 6(a), masking high-probability tokens leads to a significant degradation in the performance of the baseline GRPO. Therefore, reducing the influence of low-probability tokens on high-probability ones holds great potential for advancing RL training, as anticipated.

- **The update order is the key for *Lopti*.** The intuition behind *Lopti*, as introduced in Section 4.2, stems from the *low-probability dominant effect* of incorrectly reduced positive high-probability tokens. To confirm this intuition and rule out the possibility of random gains, we reverse the update order by processing high-probability tokens first, followed by low-probability tokens, as shown in Figure 6(b). This modification leads to significantly worse performance compared to the GRPO baseline, with training even collapsing after the 4th epoch.

- **Proper hyperparameter tuning is essential for *Advantage Reweighting* and *Lopti*.** As introduced in Section 4.2, *Advantage Reweighting* involves the hyperparameter $\alpha$, while *Lopti* depends on the hyperparameter $\eta$. For the K&K Logic Puzzles dataset, the recommended ranges are $\alpha \in [0.2, 0.3]$ and $\eta \in [0.3, 0.5]$, as values outside these ranges may result in inferior performance compared to the GRPO baseline. It is worth noting that the hyperparameter setting for *Advantage Reweighting* is task-sensitive, whereas *Lopti* demonstrates greater robustness in this regard. For math-related datasets, the optimal hyperparameter for *Advantage Reweighting* is $\alpha = 0.1$, while *Lopti* maintains its robustness with $\eta = 0.5$.

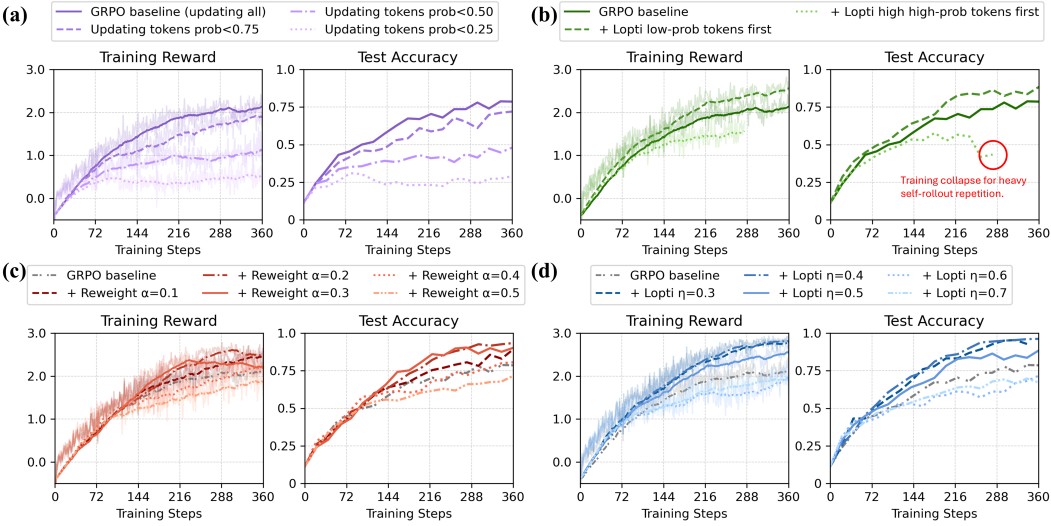

Figure 6: Ablation studies on the K&K Logic Puzzles dataset. (a) Effect of restricting updates to high-probability tokens. (b) Effect of the token update order in *Lopti*. (c) Effect of the hyperparameter $\alpha$ in *Advantage Reweighting*. (d) Effect of the hyperparameter $\eta$ in *Lopti*.

## 6 CONCLUSION

In this paper, we identify a crucial issue in RL training for LLMs: the over-dominance of low-probability tokens in model updates due to their disproportionately large gradient magnitudes. We substantiate this issue through both empirical observations and rigorous theoretical analysis. To address this imbalance, we propose two novel approaches: *Advantage Reweighting* and *Lopti*. These methods effectively mitigate gradient disparities by diminishing the undue influence of low-probability tokens, thereby facilitating more balanced and efficient updates for high-probability tokens. Extensive experiments demonstrate the effectiveness of these approaches, showing consistent improvements in GRPO-trained LLMs across diverse base models and datasets.

## ACKNOWLEDGMENTS

This work was supported in part by the General Research Fund (GRF) project 14200720 of the Hong Kong University Grants Committee. This work was also partially supported by Microsoft Research.

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

# A    THEORETICAL INTERPRETATIONS

## A.1    GRADIENT DERIVATION FOR THE GRPO OBJECTIVE

For clarity, we re-state the objective function of GRPO below:

$$J_{GRPO}(\theta) = \mathbb{E}_{q \sim \mathcal{D}, \{o_i\}_{i=1}^{G} \sim \pi_{old}}$$

$$\frac{1}{\sum_{i=1}^{G} |o_i|} \sum_{i=1}^{G} \sum_{t=1}^{|o_i|} \left\{ \underbrace{\min \left[ r_{i,t}(\theta) \hat{A}_i, \mathrm{clip}(r_{i,t}(\theta); 1 - \epsilon_l, 1 + \epsilon_h) \hat{A}_i \right]}_{J_{policy}(\theta)} - \underbrace{\beta \, \mathbb{D}_{\mathrm{KL}} \left[ \pi_\theta \| \pi_{ref} \right]}_{J_{KL}(\theta)} \right\} \quad (5)$$

$$\text{with } r_{i,t}(\theta) = \frac{\pi_\theta(o_{i,t})}{\pi_{old}(o_{i,t})}, \text{ and } \mathbb{D}_{\mathrm{KL}} \left[ \pi_\theta \| \pi_{ref} \right] = \frac{\pi_{ref}(o_{i,t})}{\pi_\theta(o_{i,t})} - \log \frac{\pi_{ref}(o_{i,t})}{\pi_\theta(o_{i,t})} - 1.$$

We begin by analyzing the policy loss term $J_{policy}(\theta)$, which originates from the PPO clipping mechanism (Schulman et al., 2017). Note that for samples with positive advantage estimates (i.e., $\hat{A}_i > 0$), the clipping is activated only when $r_{i,t}(\theta) > 1 + \epsilon_h$. Conversely, for samples with negative advantage estimates (i.e., $\hat{A}_i < 0$), the clipping becomes active only when $r_{i,t}(\theta) < 1 + \epsilon_l$. Consequently, when clipping is active, the gradient $\nabla_\theta J_{policy}(\theta)$ is zero; otherwise, it simplifies to $\nabla_\theta r_{i,t}(\theta) \cdot \hat{A}_i$. In summary, we can express the gradient of $J_{policy}(\theta)$ as

$$\nabla_\theta J_{policy}(\theta) = \frac{\nabla_\theta \pi_\theta(o_{i,t})}{\pi_{old}(o_{i,t})} \cdot \hat{A}_i \cdot \mathbb{I}_{\mathrm{trust}}\left(\frac{\pi_\theta(o_{i,t})}{\pi_{old}(o_{i,t})}, \hat{A}_i\right)$$

$$= \frac{\pi_\theta(o_{i,t})}{\pi_{old}(o_{i,t})} \cdot \hat{A}_i \cdot \mathbb{I}_{\mathrm{trust}}\left(\frac{\pi_\theta(o_{i,t})}{\pi_{old}(o_{i,t})}, \hat{A}_i\right) \nabla_\theta \log \pi_\theta(o_{i,t}) \quad (6)$$

$$\text{where } \mathbb{I}_{\mathrm{trust}}\left(\frac{\pi_\theta(o_{i,t})}{\pi_{old}(o_{i,t})}, \hat{A}_i\right) = \begin{cases} 0 & \begin{cases} \text{if } \hat{A}_i > 0 \text{ and } \frac{\pi_\theta(o_{i,t})}{\pi_{old}(o_{i,t})} > 1 + \epsilon_h \\ \text{if } \hat{A}_i < 0 \text{ and } \frac{\pi_\theta(o_{i,t})}{\pi_{old}(o_{i,t})} < 1 - \epsilon_l \end{cases} \\ 1 & \text{otherwise} \end{cases} .$$

Next, we consider the KL constraint term $J_{KL}(\theta)$, commonly referred to as $k_3$ estimation (Schulman, 2020). It provides an unbiased estimate of the KL divergence between the current policy and the reference policy. The gradient of $J_{KL}(\theta)$ is given by:

$$\nabla_\theta J_{KL}(\theta) = \beta \nabla_\theta \frac{\pi_{ref}(o_{i,t})}{\pi_\theta(o_{i,t})} + \beta \nabla_\theta \log \pi_\theta(o_{i,t})$$

$$= -\beta \frac{\pi_{ref}(o_{i,t})}{\pi_\theta(o_{i,t})^2} \nabla_\theta \pi_\theta(o_{i,t}) + \beta \nabla_\theta \log \pi_\theta(o_{i,t}) \quad (7)$$

$$= -\left[ \beta \frac{\pi_{ref}(o_{i,t})}{\pi_\theta(o_{i,t})} - \beta \right] \nabla_\theta \log \pi_\theta(o_{i,t}).$$

By combining  equation 6 and equation 7, we finally obtain the gradient of GRPO objective in the following form.

$$\nabla_\theta J_{GRPO}(\theta) = \mathbb{E}_{q \sim \mathcal{D}, \{o_i\}_{i=1}^{G} \sim \pi_{old}} \frac{1}{\sum_{i=1}^{G} |o_i|} \sum_{i=1}^{G} \sum_{t=1}^{|o_i|}$$

$$\underbrace{\left[ \frac{\pi_\theta(o_{i,t})}{\pi_{old}(o_{i,t})} \hat{A}_i \cdot \mathbb{I}_{\mathrm{trust}}\left(\frac{\pi_\theta(o_{i,t})}{\pi_{old}(o_{i,t})}, \hat{A}_i\right) + \beta \frac{\pi_{ref}(o_{i,t})}{\pi_\theta(o_{i,t})} - \beta \right]}_{w_{i,t}} \cdot \nabla_\theta \log \pi_\theta(o_{i,t}), \quad (8)$$

$$\text{where } \mathbb{I}_{\mathrm{trust}}\left(\frac{\pi_\theta(o_{i,t})}{\pi_{old}(o_{i,t})}, \hat{A}_i\right) = \begin{cases} 0 & \begin{cases} \text{if } \hat{A}_i > 0 \text{ and } \frac{\pi_\theta(o_{i,t})}{\pi_{old}(o_{i,t})} > 1 + \epsilon_h \\ \text{if } \hat{A}_i < 0 \text{ and } \frac{\pi_\theta(o_{i,t})}{\pi_{old}(o_{i,t})} < 1 - \epsilon_l \end{cases} \\ 1 & \text{otherwise} \end{cases} .$$

## A.2 Proof for Proposition 4.2

**Proof.** As introduced in Section 4.1, we denote LLM as a composite function $f = f_L \circ f_{L-1} \circ \cdots \circ f_1$, where each $f_\ell$ (with $\ell \in \{1, \ldots, L\}$) corresponds to a distinct layer of the network. $\boldsymbol{a}_{\ell-1}$ denotes the input and $\boldsymbol{a}_\ell$ denotes the output of $\ell$th layer, and the Jacobian matrix of the $\ell$th layer with respect to its input is expressed as $J_\ell := \frac{\partial f_\ell(\boldsymbol{a}_{\ell-1})}{\partial \boldsymbol{a}_{\ell-1}}$. For any token $o_{i,t}$, we denote the gradient of GRPO objective with respect to the activations $\boldsymbol{a}_\ell$ at $\ell$th layer as $\boldsymbol{\delta}_\ell(o_{i,t}) := \nabla_{\boldsymbol{a}_\ell} J_{GRPO}(o_{i,t})$. According to the rule of backpropagation, we have:

$$\boldsymbol{\delta}_\ell(o_{i,t}) = J_{\ell+1}^\mathsf{T} \boldsymbol{\delta}_{\ell+1}(o_{i,t}) = \prod_{j=\ell+1}^{L} J_j^\mathsf{T} \cdot \boldsymbol{\delta}_L(o_{i,t}). \tag{9}$$

Note that the gradients of all intermediate layers are back-propagated from the last layer of LLM, thereby we discuss the gradients of the last layer ($\boldsymbol{\delta}_L(o_{i,t})$) first. The last-layer output of an LLM is the logits $\boldsymbol{a}_L = (a_L^1, a_L^2, \ldots, a_L^N)$, which corresponds to a finite vocabulary $\mathcal{V} = \{v^1, v^2, \ldots, v^N\}$. The output probability of the corresponding token is calculated through softmax operation:

$$\pi_\theta(v^n) = \frac{e^{a_L^n}}{\sum_{m=1}^{N} e^{a_L^m}}, \quad \text{for } \forall n \in \{1, 2, \ldots, N\}. \tag{10}$$

Given a token $o_{i,t}$, let $k$ denote the index of the logits head corresponding to this token (i.e., $v^k = o_{i,t}$). To obtain the gradient of last layer of LLM, we have:

$$\begin{aligned}
\frac{\partial J_{GRPO}(o_{i,t})}{\partial a_L^n} &\overset{i}{=} w_{i,t} \cdot \frac{\partial \log \pi_\theta(o_{i,t})}{\partial a_L^n} \\
&\overset{ii}{=} w_{i,t} \cdot \sum_{m=1}^{N} \frac{\partial \log \pi_\theta(o_{i,t})}{\partial \pi_\theta(v^m)} \cdot \frac{\partial \pi_\theta(v^m)}{\partial a_L^n} \\
&\overset{iii}{=} w_{i,t} \cdot \frac{\partial \log \pi_\theta(o_{i,t})}{\partial \pi_\theta(v^k)} \cdot \frac{\partial \pi_\theta(v^k)}{\partial a_L^n} = w_{i,t} \cdot \frac{1}{\pi_\theta(v^k)} \cdot \frac{\partial \pi_\theta(v^k)}{\partial a_L^n}.
\end{aligned} \tag{11}$$

Here, equality (i) follows from equation 8; equality (ii) is obtained by applying the chain rule during backpropagation; and equality (iii) holds because $\partial \log \pi_\theta(o_{i,t})/\pi_\theta(v^m) = 0$ for all $m \neq k$. Next, we consider the following two cases for the gradient on the logits head $a_L^n$ ($n \in \{1, 2, \ldots, N\}$).

**Case 1: the logits head corresponds to the sampled token ($n = k$)**

$$\begin{aligned}
\frac{\partial J_{GRPO}(o_{i,t})}{\partial a_L^k} &= w_{i,t} \cdot \frac{1}{\pi_\theta(v^k)} \cdot \frac{\partial \pi_\theta(v^k)}{\partial a_L^k} \\
&= w_{i,t} \cdot \frac{1}{\pi_\theta(v^k)} \cdot \frac{e^{a_L^n} \cdot \sum_{m=1}^{N} e^{a_L^m} - e^{2a_L^n}}{(\sum_{m=1}^{N} e^{a_L^m})^2} \\
&= w_{i,t} \cdot \frac{1}{\pi_\theta(v^k)} \cdot \pi_\theta(v^k) \cdot (1 - \pi_\theta(v^k)) \\
&= w_{i,t} \cdot (1 - \pi_\theta(v^k)).
\end{aligned} \tag{12}$$

**Case 2: the logits head corresponds to the un-sampled token ($n \neq k$)**

$$\begin{aligned}
\frac{\partial J_{GRPO}(o_{i,t})}{\partial a_L^n} &= w_{i,t} \cdot \frac{1}{\pi_\theta(v^k)} \cdot \frac{\partial \pi_\theta(v^k)}{\partial a_L^n} \\
&= w_{i,t} \cdot \frac{1}{\pi_\theta(v^k)} \cdot \frac{-e^{a_L^k} \cdot e^{a_L^n}}{(\sum_{m=1}^{N} e^{a_L^m})^2} \\
&= w_{i,t} \cdot \frac{1}{\pi_\theta(v^k)} \cdot \pi_\theta(v^k) \cdot (-\pi_\theta(v^n)) \\
&= w_{i,t} \cdot (-\pi_\theta(v^n)).
\end{aligned} \tag{13}$$

For simplicity, we denote the vector distribution output across the vocabulary as $\boldsymbol{p}$, and denote $\boldsymbol{I}(o_{i,t})$ as the one-hot vector with its only non-zero component at $k$th position (i.e., the position correspondence to token $o_{i,t}$). We have the following expressions

$$
\begin{aligned}
\boldsymbol{p}(o_{i,t}) &= (\pi_\theta(v^1), \pi_\theta(v^2), \ldots, \pi_\theta(v^N)) \in \mathcal{R}^N \\
\boldsymbol{I}(o_{i,t}) &= (0, 0, \ldots, \underbrace{1}_{k\text{th}}, \ldots, 0) \in \mathcal{R}^N.
\end{aligned}
\tag{14}
$$

Combining equation 12 and equation 13, and utilizing the notation defined in equation 14, we obtain:

$$
\boldsymbol{\delta}_L(o_{i,t}) = \nabla_{\boldsymbol{a}_L} J_{GRPO}(o_{i,t}) = w_{i,t} \cdot (\boldsymbol{I}(o_{i,t}) - \boldsymbol{p}(o_{i,t})).
\tag{15}
$$

Considering the lower bound for the gradient norm, we have:

$$
\begin{aligned}
\|\boldsymbol{\delta}_L(o_{i,t})\| &= |w_{i,t}| \cdot \|\boldsymbol{p}(o_{i,t}) - \boldsymbol{I}(o_{i,t})\| \\
&= |w_{i,t}| \cdot \sqrt{(1 - \pi_\theta(v^k))^2 + \sum\nolimits_{n \neq k}^N \pi_\theta(v^n)^2} \\
&\geq |w_{i,t}| \cdot \sqrt{(1 - \pi_\theta(v^k))^2 + \frac{1}{N-1}\left(\sum\nolimits_{n \neq k}^N \pi_\theta(v^n)\right)^2} \\
&= |w_{i,t}| \cdot \sqrt{(1 - \pi_\theta(v^k))^2 + \frac{1}{N-1}(1 - \pi_\theta(v^k))^2} \\
&= |w_{i,t}| \cdot \sqrt{\frac{N}{N-1}}(1 - \pi_\theta(o_{i,t})),
\end{aligned}
\tag{16}
$$

where the inequality follows from the Cauchy-Schwarz inequality. The equality holds holding if and only if $\pi_\theta(v^n)$ is uniformly distributed for all $n \neq k$.

By substituting equation 16 into equation 9, we obtain:

$$
\begin{aligned}
\|\boldsymbol{\delta}_\ell(o_{i,t})\| &= \|\prod\nolimits_{j=\ell+1}^L J_j^\mathsf{T} \cdot \boldsymbol{\delta}_L(o_{i,t})\| \\
&\overset{i}{\geq} \prod\nolimits_{j=\ell+1}^L \sigma_{\min}(J_j^\mathsf{T}) \cdot \|\boldsymbol{\delta}_L(o_{i,t})\| \\
&\overset{ii}{\geq} \prod\nolimits_{j=\ell+1}^L c_j \cdot \|\boldsymbol{\delta}_L(o_{i,t})\| \\
&\overset{iii}{\geq} \prod\nolimits_{j=\ell+1}^L c_j \cdot |w_{i,t}| \cdot \sqrt{\frac{N}{N-1}}(1 - \pi_\theta(v^k)),
\end{aligned}
\tag{17}
$$

where inequality (i) follows from the variational characterization of singular values, inequality (ii) is a consequence of Assumption 4.1, and inequality (iii) results from equation 16.

Next, considering an alternative direction, we derive an upper bound for the gradient norm:

$$
\begin{aligned}
\|\boldsymbol{\delta}_L(o_{i,t})\| &= |w_{i,t}| \cdot \sqrt{(1 - \pi_\theta(v^k))^2 + \sum\nolimits_{n \neq k}^N \pi_\theta(v^n)^2} \\
&\leq |w_{i,t}| \cdot \sqrt{(1 - \pi_\theta(v^k))^2 + \sum\nolimits_{n \neq k}^N \pi_\theta(v^n)^2 + 2\sum\nolimits_{n,m \neq k, n<m}^N \pi_\theta(v^n)\pi_\theta(v^m)} \\
&= |w_{i,t}| \cdot \sqrt{(1 - \pi_\theta(v^k))^2 + \left(\sum\nolimits_{n \neq k}^N \pi_\theta(v^n)\right)^2} \\
&= |w_{i,t}| \cdot \sqrt{(1 - \pi_\theta(v^k))^2 + (1 - \pi_\theta(v^k))^2} \\
&= |w_{i,t}| \cdot \sqrt{2}(1 - \pi_\theta(o_{i,t})),
\end{aligned}
\tag{18}
$$

where the inequality holds because $\pi_\theta(v^n) \geq 0$ for all $n \in 1, 2, \ldots, N$. The equality is achieved if and only if there exists an index $m$ such that $\pi_\theta(v^m) = 1 - \pi_\theta(v^k)$ and $\pi_\theta(v^m) = 0$ for all $n \neq m$ and $n \neq k$.

Similarly, substituting equation 18 into equation 9, we have

$$
\begin{aligned}
\|\boldsymbol{\delta}_\ell(o_{i,t})\| &= \|\prod\nolimits_{j=\ell+1}^{L} J_j^\mathsf{T} \cdot \boldsymbol{\delta}_L(o_{i,t})\| \\
&\leq \prod\nolimits_{j=\ell+1}^{L} \sigma_{\max}(J_j^\mathsf{T}) \cdot \|\boldsymbol{\delta}_L(o_{i,t})\| \\
&\leq \prod\nolimits_{j=\ell+1}^{L} d_j \cdot \|\boldsymbol{\delta}_L(o_{i,t})\| \\
&\leq \prod\nolimits_{j=\ell+1}^{L} d_j \cdot |w_{i,t}| \cdot \sqrt{2}\left(1 - \pi_\theta(v^k)\right),
\end{aligned}
\tag{19}
$$

where the inequalities hold for the same reasons as in equation 17. Together, equation 17 and equation 19 establish the result of Proposition 4.2.

## B  HYPERPARAMETER SETTINGS

As described in Section 4.2, our proposed *Advantage Reweighting* and *Lopti* require only minor modifications to the existing GRPO training framework. Our implementation is built upon the `verl` library[*] (Sheng et al., 2025). The key hyperparameter configurations for GRPO training are detailed in Table 2. Note that we adopt the 'clip higher' technique from DAPO (Yu et al., 2025) to stabilize entropy and mitigate entropy collapse. All other hyperparameters adhere to the default settings provided by `verl`.

The hyperparameter configurations specific to *Advantage Reweighting* and *Lopti* are summarized in Table 3. As reported in Section 5, while the joint application of the two techniques generally yields improved results for the K&K Logic Puzzle dataset, this is not the case for the Math dataset. Please refer to Appendix C.3 for a detailed explanation. Consequently, using either technique individually is recommended for the math-related dataset.

It should be noted that the hyperparameter settings for our proposed methods are related to the task specification, but not to the base model utilized. As we concluded in Appendix C.3, for reasoning tasks where the model uses natural language for inference, low-probability tokens occur more frequently in the sampled batch, and therefore their dominant effect is more pronounced. Consequently, integrating *Advantage Reweighting* and *Lopti* can yield better results. On the other hand, for tasks such as mathematics, where the model uses specialized mathematical notation for inference, low-probability tokens occur less frequently. The dominant effect is not as pronounced under such circumstances. Regarding the hyperparameter $\alpha$ for *Advantage Reweighting*, for tasks where low-probability tokens exhibit a greater dominant effect (such as reasoning tasks), it should be set to 0.2–0.3 to achieve the best performance. For tasks where the dominant effect is weaker (such as math tasks), it should be set to 0.1 to achieve the best performance. As for the hyperparameter $\eta$ for *Lopti*, it is robust across tasks, and setting the value to 0.5 is sufficient for all tasks.

For consistency, the same seed is used across all experiments. We save a checkpoint every 20 RL steps, and all evaluation accuracies reported on the test set in this paper are averaged over the last three checkpoints. The detailed implementation can be found in our code[†].

## C  EXPERIMENTAL DETAILS

### C.1  TASK DESCRIPTION

#### C.1.1  K&K LOGIC PUZZLE

As introduced in Section 5.1, the K&K logic puzzles involve a fictional scenario where inhabitants of an island are either Knights, who always tell the truth, or Knaves, who always lie. The objective of the LLMs is to determine the identity of each inhabitant (Knight or Knave) based on a set of statements they make about themselves and others. Following Logic-RL (Xie et al., 2025), we utilize the LLMs

---

[*]https://github.com/volcengine/verl
[†]We provide the source code in supplementary materials.

Table 2: Key hyperparameters for GRPO training, with the corresponding variable names in the `verl` configuration indicated in brackets.

| Hyperparameter | Value | |
| --- | --- | --- |
| | K&K | Math |
| **Rollout-related** | | |
| Sampling temperature (`temperature`) | 0.7 | 1.0 |
| Question num per batch (`ppo_mini_batch_size`) | 64 | 128 |
| Answer num per question (`rollout.n`) | 8 | |
| Max tokens num per response (`max_response_length`) | 4096 | |
| **Training-related** | | |
| Update batch size (`ppo_micro_batch_size`) | 256 | 512 |
| Optimizer (`optim.type`) | adamw | |
| Learning rate (`optim.lr`) | 1e-6 | |
| KL divergence coefficient (`kl_loss_coef`) | 0.001 | |
| Lower clipping threshold (`clip_ratio_low`) | 0.2 | |
| Upper clipping threshold (`clip_ratio_high`) | 0.24 | |

Table 3: Hyperparameter settings for *Advantage Reweighting* and *Lopti*.

| Hyperparameter | Value | |
| --- | --- | --- |
| | K&K | Math |
| Advantage Reweighting ($\alpha$) | 0.3 | 0.1 |
| Lopti ($\eta$) | 0.5 | 0.5 |
| Joint operation for better results | True | False |

after instruction fine-tuning (`Qwen2.5-3B-Instruct` and `Qwen2.5-7B-Instruct-1M`) as starting point. The prompt specifically designed for the LLMs is as follows.

> **Prompt**
>
> system\n You are a helpful assistant. The assistant first thinks about the reasoning process in the mind and then provides the user with the answer. The reasoning process and answer are enclosed within <think> </think> and<answer> </answer> tags, respectively, i.e., <think> reasoning process here </think><answer> answer here </answer>. Now the user asks you to solve a logical reasoning problem. After thinking, when you finally reach a conclusion, clearly state the identity of each character within <answer> </answer> tags. i.e., <answer> (1) Zoey is a knight\n (2) ... </answer>.\n \n user\n {problem}\n \n assistant\n <think>

To encourage LLMs to exhibit chain-of-thought (CoT) reasoning, Logic-RL (Xie et al., 2025) designs a reward function consisting of two components, as outlined in Table 4. The output format is deemed completely correct if LLMs include CoT reasoning enclosed within `<think></think>` tags and the final answer enclosed within `<answer></answer>` tags.

Table 4: Reward design for K&K Logic Puzzle proposed in Logic-RL (Xie et al., 2025)

| | Format Reward | Answer Reward |
| --- | --- | --- |
| Completely Correct | 1 | 2 |
| Patially Correct | -1 | -1.5 |
| Completely Wrong | -1 | -2 |

For the K&K Logic Puzzle dataset, the number of players (ranging from 3 to 7) can be adjusted to control the difficulty level, with a greater number of players resulting in higher difficulty. To provide an intuitive illustration, we present an easy example with 3 players and a challenging example with 7 players below. Without utilizing curriculum learning, we directly train the LLMs on the mixed training set for a total of 5 epochs.

---

**An Example of K&K Puzzle with 3 people**

**Problem:**
A very special island is inhabited only by knights and knaves. Knights always tell the truth, and knaves always lie. You meet 3 inhabitants: Alexander, Lily, and Samuel. Alexander remarked, "Lily is a knave or Lily is a knight". In a statement by Lily: "Samuel is a knight if and only if Lily is a knight". Samuel was heard saying, "Lily is a knight". So who is a knight and who is a knave?

**Example Reasoning Process:**
• Assume Alexander is a knight. No contradiction is found in their claim that Lily is a knave or Lily is a knight.
• Assume Lily is a knight. No contradiction is found in their claim that Samuel is a knight if and only if Lily is a knight.
• Assume Samuel is a knight. No contradiction is found in their claim that Lily is a knight.

**Standard Solution:**
(1) Alexander is a knight, (2) Lily is a knight, (3) Samuel is a knight

---

**An Example of K&K Puzzle with 7 people**

**Problem:**
A very special island is inhabited only by knights and knaves. Knights always tell the truth, and knaves always lie. You meet 7 inhabitants: Harper, Emma, Mia, Luke, Alexander, David, and Ethan. As Harper put it, "David is not a knight". In Emma's words: "David is a knight". Mia said that If Emma is a knight then Emma is a knave. Luke said, "If Alexander is a knave then Emma is a knight." Alexander was heard saying, "If David is a knight then Harper is a knave". "Alexander is not a knight" - David. "Harper is a knight," Ethan mentioned. So who is a knight and who is a knave?

**Example Reasoning Process:**
• Assume Harper is a knight. No contradiction is found in their claim that David is not a knight.
• David cannot be a knight, because this would contradict the claim of Harper that David is not a knight.
• Assume David is a knave. No contradiction is found in their false claim that Alexander is not a knight.
• Assume Alexander is a knight. No contradiction is found in their claim that If David is a knight then Harper is a knave.
• Emma cannot be a knight, because this would contradict the claim of their own that David is a knight.
• Assume Emma is a knave. No contradiction is found in their false claim that David is a knight.
• Assume Mia is a knight. No contradiction is found in their claim that If Emma is a knight then Emma is a knave.
• Assume Luke is a knight. No contradiction is found in their claim that If Alexander is a knave then Emma is a knight.
• Assume Ethan is a knight. No contradiction is found in their claim that Harper is a knight.

**Standard Solution:**
(1) Harper is a knight (2) Emma is a knave (3) Mia is a knight (4) Luke is a knight (5) Alexander is a knight (6) David is a knave (7) Ethan is a knight

---

The detailed training records for `Qwen2.5-3B-Instruct` and `Qwen2.5-7B-Instruct-1M` are presented in Figure 7 and Figure 8, respectively. In addition to the points discussed in Section 5.1, it is worth noting that our *Advantage Reweighting* and *Lopti* approaches slightly increase the response length while significantly reducing the gradient norm compared to the naive GRPO. Both observations empirically suggest that the RL training process is further stabilized.

For the six categories of inference-related words used in the linguistic analysis, the detailed word lists are provided in Table 5. It is important to note that for the nouns and verbs listed in the table, their conjugated forms are also included in the analysis. Specifically, we account for the plural forms of nouns as well as the past tense and past participle forms of verbs. Additionally, uppercase and lowercase letters are treated equivalently.

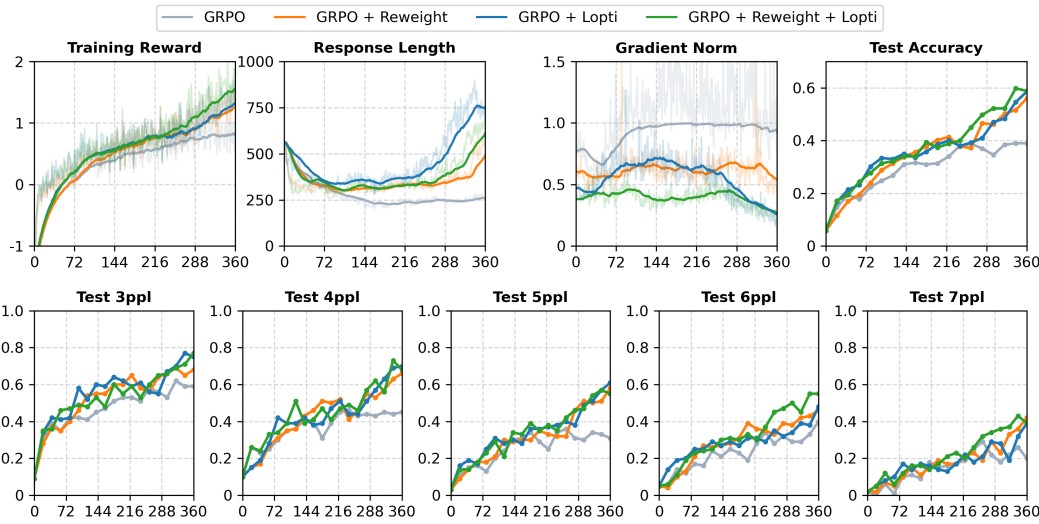

Figure 7: Experimental records of `Qwen2.5-3B-Instruct` trained with GRPO on the K&K Logic Puzzle dataset. The training curve is smoothed through exponential moving average with coefficient of 0.95.

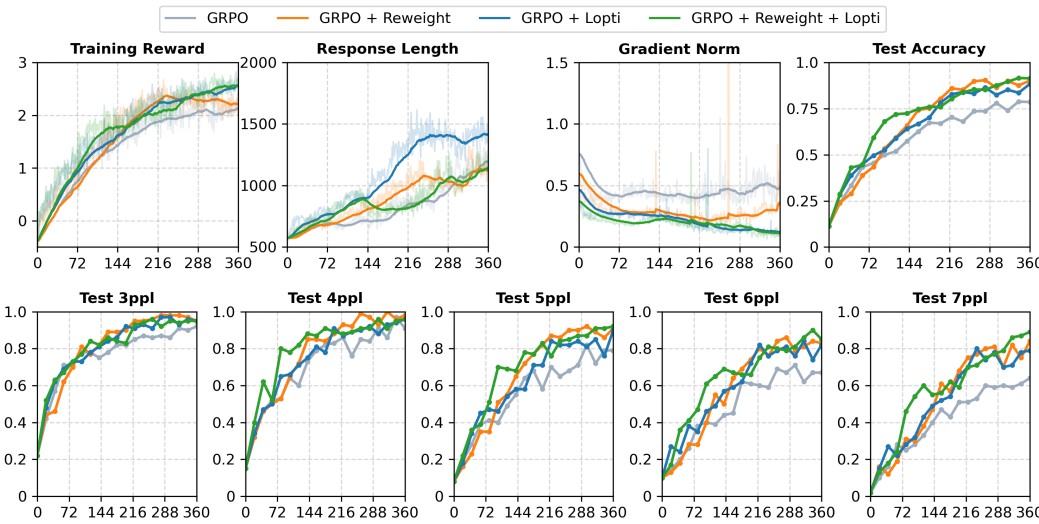

Figure 8: Experimental records of `Qwen2.5-7B-Instruct-1M` trained with GRPO on the K&K Logic Puzzle dataset.

### C.1.2  MATH DATASET

As discussed in Section 5.2, we perform additional experiments on two math-related datasets, DSR-Uniform and ORZ. Consistent with the majority of prior studies, we use `Qwen2.5-7B` as the starting point. It is important to note that `Qwen2.5-7B` undergoes no post-training. This setup is therefore referred to as a "cold-start" and denoted as RL-Zero (Guo et al., 2025). No instruction-following templates are employed; instead, we use the following straightforward prompt.

Table 5: Six categories of inference-related words associated with LLMs' performance on the K&K Logic Puzzles dataset.

| Category | Words (Nouns and verbs include their conjugated forms) |
| --- | --- |
| Analysis | 'analyze', 'consider', 'look at', 'check', 'examine' |
| Statement | XXX's 'statement' |
| Causal Indicator | 'since', 'because', 'due to', 'given that' |
| Conclusion Indicator | 'so', 'thus', 'hence', 'as a result', 'consequently', 'therefore' |
| Assumption | 'assume', 'if...then...' |
| Assertion | 'must be', 'definite' |

**Prompt**

{problem} Let's think step by step and output the final answer within \\boxed{}.

LLMs that have not undergone post-training typically exhibit poor performance in adhering to specific output formats. As a result, format-related points were not included during training. Additionally, math problems are generally not partially correct, making a binary reward sufficient for evaluating the LLMs' output. Specifically, a reward of 1 is assigned when LLMs produce the correct answer, and 0 otherwise.

The detailed experimental results for the DSR-Uniform and ORZ datasets are presented in Figure 9 and Figure 10, respectively. Notably, the training curve for DSR-Uniform demonstrates a continual learning trend, with the reward progressively increasing over time. In contrast, this is not observed for ORZ, where the reward converges rapidly within 100 steps. However, the test accuracy curves for both datasets converge to a stable value within 100 steps, after which they exhibit only minor fluctuations. Despite these patterns, the improvements achieved by our proposed methods, *Advantage Reweighting* and *Lopti*, remain clearly observable.

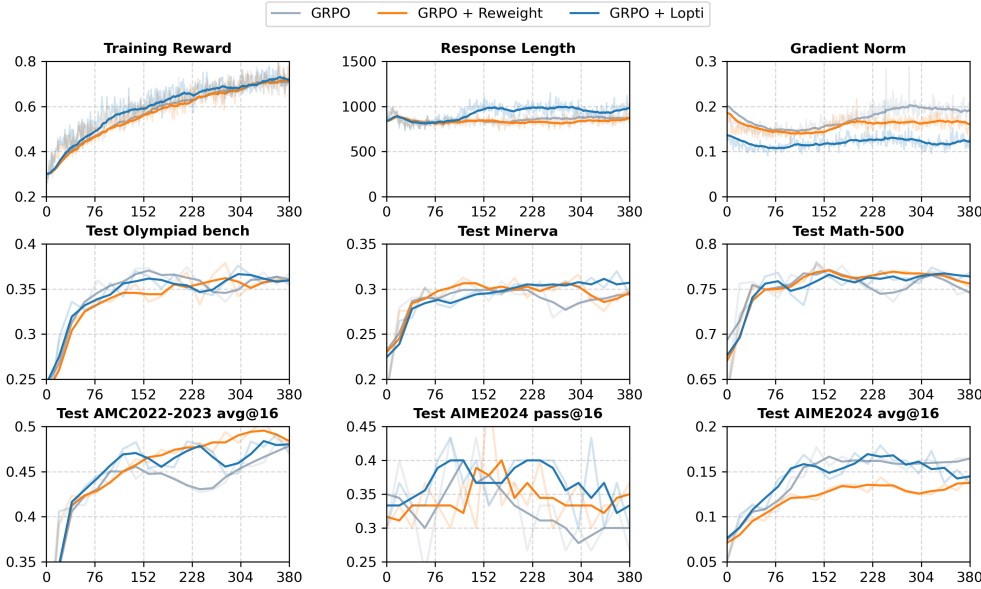

Figure 9: Experimental records of `Qwen2.5-7B` trained with GRPO on DSR-uniform dataset. The training curve is smoothed through exponential moving average with coefficient of 0.95, and the testing curve is smoothed with a window size of 3.

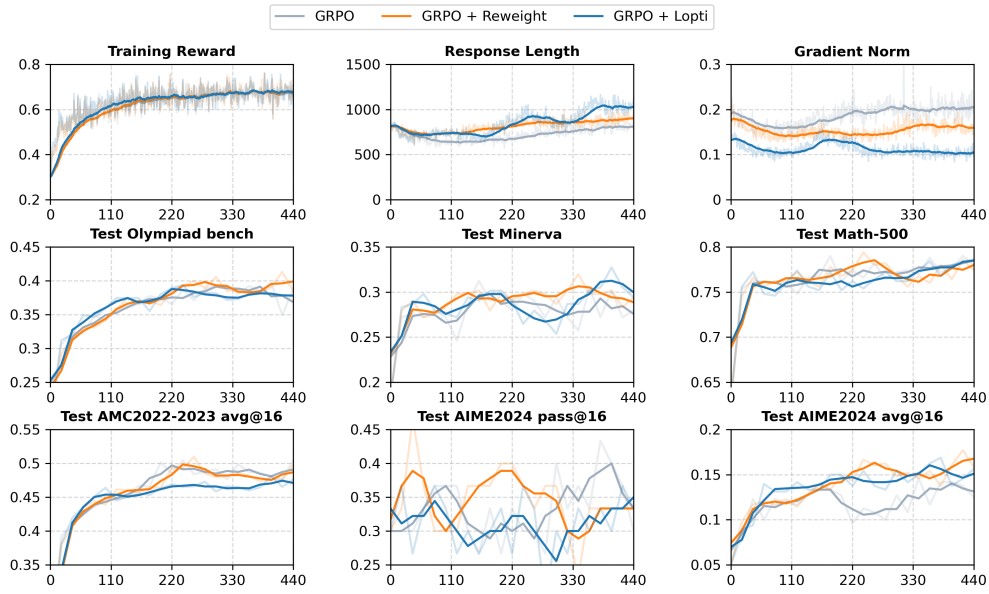

Figure 10: Experimental records of `Qwen2.5-7B` trained with GRPO on ORZ dataset.

## C.2 COMPUTATIONAL COSTS

Our experiments are conducted on a single machine equipped with an AMD EPYC 7V13 64-Core CPU and four NVIDIA A100 80GB PCIe GPUs. The experiments on the K&K Logic Puzzle dataset require approximately 16–22 hours to complete (excluding testing during the training process), while those on the math-related dataset take around 37–48 hours.

The *Advantage Reweighting* involves only recalculating the advantage of tokens, with a time overhead in the range of milliseconds. However, this efficiency does not apply to *Lopti*, as it splits the tokens in a batch into two groups and performs updates twice. Consequently, the updating process requires twice the amount of time, as detailed in Table 6.

Table 6: Computational cost comparison of *Lopti* operation over the first 50 training steps on K&K Logic Puzzle Dataset.

| | Time (s)/step | | | | | |
| Procedure | Deepseek-Distill-1.5B | | Qwen2.5-3B-Instruct | | Qwen2.5-7B-Instruct-1M | |
| | w/o Lopti | w/ Lopti | w/o Lopti | w/ Lopti | w/o Lopti | w/ Lopti |
|---|---|---|---|---|---|---|
| **Sampling** | 140.9 | 141.7 | 25.4 | 27.8 | 68.5 | 69.3 |
| **Training** | 100.8 | 179.2 | 17.6 | 35.3 | 61.4 | 116.8 |
| **Others** | 10.1 | 9.4 | 2.4 | 2.8 | 10.3 | 10.2 |
| **Total** | 251.8 | 330.3 | 45.4 | 65.9 | 140.2 | 196.3 |

## C.3 INCOMPATIBILITY OF AR AND LOPTI FOR SIMULTANEOUS APPLICATION TO MATH-RELATED DATASETS

As reported in Section 5, although the joint application of the two techniques (*Advantage Reweighting* and *Lopti*) generally yields improved results on the K&K Logic Puzzle dataset, this is not observed for the Math dataset. To investigate the underlying cause of this discrepancy, we perform an analysis on the DeepScaleR-Uniform dataset analogous to that in Figure 1, and present a comparison between the K&K Logic Puzzle and DeepScaleR-Uniform datasets in Figure 11.

When the task involves "logic puzzle", LLMs encounter a greater number of "high-entropy" positions during the auto-regressive generation process. This remains true even when we lower the temperature to 0.7 (as opposed to the typical setting of 1.0 used in mathematical tasks). As a result, the proportion

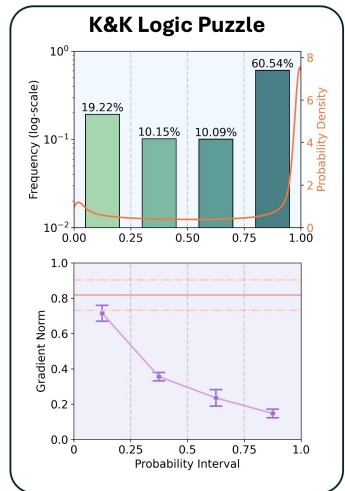 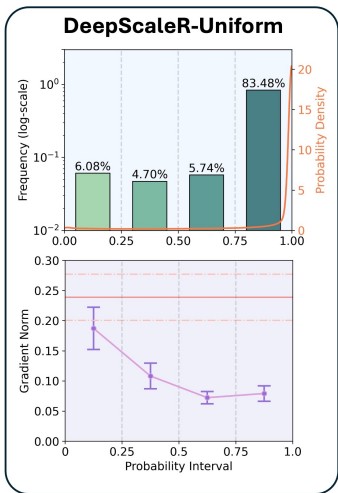

Figure 11: Comparison of token probability distributions and gradient norms during GRPO training of `Qwen2.5-7B-Instruct-1M` between the K&K Logic Puzzle dataset and the DeepScaleR-Uniform dataset. The proportion of sampled low-probability tokens during training is substantially smaller for the DeepScaleR-Uniform dataset than for the K&K Logic Puzzle dataset.

of "low-probability tokens" in the sampled batch increases. In such scenarios, the dominance effect of low-probability tokens becomes more pronounced, allowing both "Advantage Reweighting" and "Lopti" to perform effectively. When used independently, there is no significant difference in performance; when combined, they can even yield superior results.

In contrast, for math-related tasks, high-entropy positions are encountered less frequently during generation, leading to a lower proportion of low-probability tokens in the sampled batch compared to logic puzzles. This may be attributed to the greater textual inertia and determinism inherent in mathematical language. Consequently, in this setting, combining the two methods can overly suppress the contribution of low-probability tokens to the gradient updates, ultimately resulting in suboptimal performance.

## D  ADDITIONAL EXPERIMENTAL RESULTS

### D.1  ADDITIONAL POLICY-GRADIENT-BASED RL ALGORITHMS

In addition to GRPO, our proposed methods, *Advantage Reweighting* and *Lopti*, are also well-adapted to other Policy Gradient-based RL algorithms. In this section, we extend our methods to REINFORCE++ (Hu, 2025), and DAPO (Yu et al., 2025).

### D.1.1  REINFORCE++

REINFORCE++ (Hu, 2025) is a widely recognized algorithm that builds upon the conventional REINFORCE (Williams, 1992) while incorporating various stabilization techniques introduced by PPO (Schulman et al., 2017).

Similar to GRPO, REINFORCE++ also eliminates the need for a value model, thereby reducing computational costs compared to PPO. The key differences between GRPO and REINFORCE++ lie in how they *estimate the advantage* and *constrain the distance between the RL-trained model and the initial (or reference) model*. GRPO estimates the advantage based on the difference between the reward and the group-relative expected return, incorporating the KL constraint directly into the objective function (cf. Section 3 for details). In contrast, REINFORCE++ does not emphasize the concept of 'group' under the same prompt. Instead, it estimates the advantage directly from the reward and treats the KL constraint as a penalty term added to the reward. Specifically, REINFORCE++ estimates the advantage as follows:

$$\hat{A}_{i,t} = \frac{\hat{A}_{i,t}^{R++} - \mu_A}{\sigma_A} \text{ with } \hat{A}_{i,t}^{R++} = r(\boldsymbol{q}, \boldsymbol{o}_i) - \beta \cdot \sum_{j=t}^{T} \mathbb{D}_{\mathrm{KL}} \left[ \pi_\theta(o_{i,j}) \| \pi_{ref}(o_{i,j}) \right], \quad (20)$$

where $\mu_A$ and $\sigma_A$ represent the mean and standard deviation of the advantages of all tokens within the RL-sampled batch, respectively. The KL divergence term is computed using the k1 estimation (Schulman, 2020): $\mathbb{D}_{\mathrm{KL}} \left[ \pi_\theta(o_{i,j}) \| \pi_{ref}(o_{i,j}) \right] = \pi_\theta(o_{i,j}) / \pi_{ref}(o_{i,j})$. The optimization objective of REINFORCE++ is:

$$J_{R++}(\theta) = \mathbb{E}_{\boldsymbol{q} \sim \mathcal{D}, \{\boldsymbol{o}_i\}_{i=1}^{G} \sim \pi_{old}}$$

$$\frac{1}{\sum_{i=1}^{G} |\boldsymbol{o}_i|} \sum_{i=1}^{G} \sum_{t=1}^{|\boldsymbol{o}_i|} \left\{ \min \left[ r_{i,t}(\theta) \hat{A}_{i,t}, \mathrm{clip}(r_{i,t}(\theta); 1 - \epsilon_l, 1 + \epsilon_h) \hat{A}_{i,t} \right] \right\} \quad (21)$$

$$\text{with } r_{i,t}(\theta) = \frac{\pi_\theta(o_{i,t} | \boldsymbol{q}, \boldsymbol{o}_{i,<t})}{\pi_{old}(o_{i,t} | \boldsymbol{q}, \boldsymbol{o}_{i,<t})}.$$

Similar to the experiments conducted with GRPO, we validate two base models as starting points: `Qwen2.5-3B-Instruct` and `Qwen2.5-7B-Instruct-1M`. All hyperparameters of REINFORCE++ are kept consistent with those used for GRPO, as described in Appendix B. The only difference is that, on the K&K Logic Puzzle dataset, the optimal hyperparameter setting for *Advantage Reweighting* is $\alpha = 0.1$ for REINFORCE++, and $\alpha = 0.3$ for GRPO.

The evaluation results on the test set are reported in Table 7. Notably, the performance of naive REINFORCE++ is slightly worse than that of naive GRPO (cf. Figure 4). This observation aligns with the findings of Xiong et al. (2025), as the advantage normalization method in REINFORCE++ may introduce unnecessary bias toward entirely incorrect responses on overly difficult prompts. Nevertheless, the improvements achieved by our proposed methods, *Advantage Reweighting* and *Lopti*, remain significant. For more details on the training process, please refer to the records presented in Figure 12 and Figure 13.

Table 7: Experimental results of REINFORCE++ on the K&K Logic Puzzles dataset. For *Advantage Reweight*, $\alpha = 0.1$, and for *Lopti*, $\eta = 0.5$. The evaluation accuracy on the test set are averaged over the last three checkpoints to mitigate randomness.

| Model | Difficulty by Number of People | | | | | |
| | 3 | 4 | 5 | 6 | 7 | Avg. |
|---|---|---|---|---|---|---|
| **Qwen2.5-3B-Instruct** | 0.09 | 0.10 | 0.03 | 0.05 | 0.02 | 0.06 |
| REINFORCE++ | 0.37 | 0.31 | 0.20 | 0.21 | 0.06 | 0.23 |
| **REINFORCE++ with Reweight** | 0.53 | 0.44 | 0.31 | 0.26 | 0.14 | 0.34 (↑46.1%) |
| **REINFORCE++ with Lopti** | 0.47 | 0.36 | 0.26 | 0.26 | 0.12 | 0.29 (↑27.8%) |
| **REINFORCE++ with Reweight & Lopti** | 0.61 | 0.49 | 0.38 | 0.34 | 0.21 | 0.41 (↑76.5%) |
| **Qwen2.5-7B-Instruct-1M** | 0.22 | 0.15 | 0.08 | 0.10 | 0.02 | 0.11 |
| REINFORCE++ | 0.68 | 0.72 | 0.54 | 0.42 | 0.43 | 0.56 |
| **REINFORCE++ with Reweight** | 0.81 | 0.77 | 0.66 | 0.62 | 0.48 | 0.67 (↑19.7%) |
| **REINFORCE++ with Lopti** | 0.89 | 0.85 | 0.71 | 0.66 | 0.51 | 0.72 (↑29.7%) |
| **REINFORCE++ with Reweight & Lopti** | 0.87 | 0.88 | 0.81 | 0.71 | 0.69 | 0.79 (↑41.9%) |

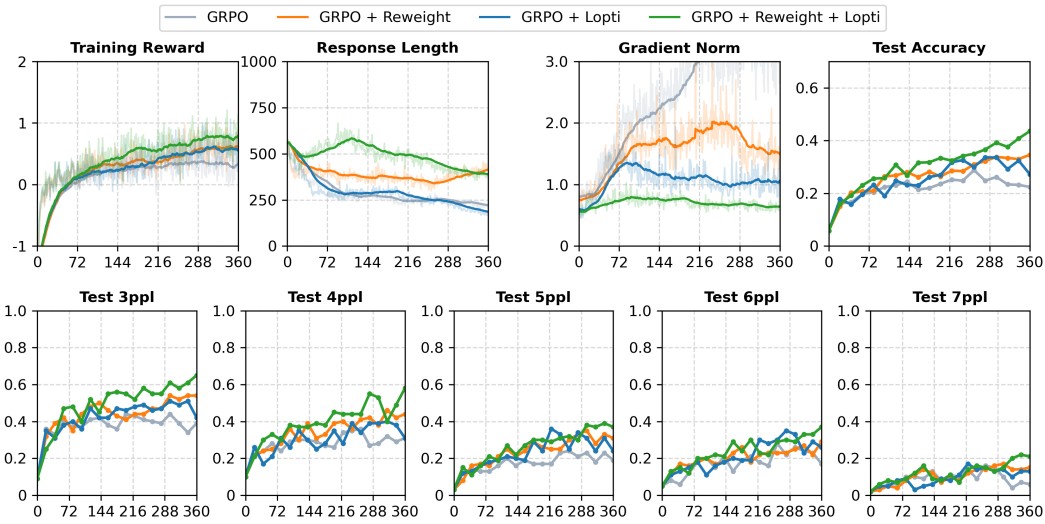

Figure 12: Experimental records of `Qwen2.5-3B-Instruct` trained with REINFORCE++ on the K&K Logic Puzzle dataset. The training curve is smoothed through exponential moving average with coefficient of 0.95.

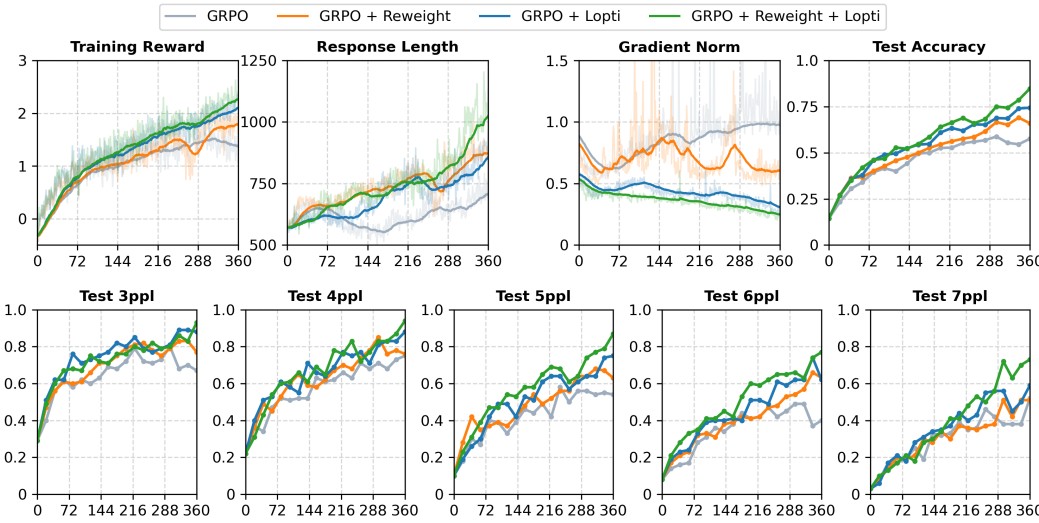

Figure 13: Experimental records of `Qwen2.5-7B-Instruct-1M` trained with REINFORCE++ on the K&K Logic Puzzle dataset.

### D.1.2 DAPO

DAPO (Yu et al., 2025) is a recently proposed algorithm that builds upon GRPO (Liu et al., 2025). Beyond the foundation of GRPO, it further introduces four key techniques to accelerate and stabilize training:

- **Clip Higher**, which establishes a higher clip threshold for PPO-style importance sampling clipping on positive samples. This proves effective in promoting system diversity and preventing entropy collapse;
- **Dynamic Sampling**, which filters out prompts that yield homogeneous responses with identical rewards, thereby improving training efficiency and stability;

- **Token-Level Policy Gradient Loss**, which averages the loss over all tokens within a batch rather than over sequences, a modification that is critical in long-chain-of-thought RL scenarios;
- **Overlong Reward Shaping**, which mitigates reward noise caused by hard truncation when responses exceed length limits, thereby reducing noise and stabilizing training.

Table 8: Experimental results of DAPO on the K&K Logic Puzzles dataset.

| Model | Difficulty by Number of People | | | | | |
| | 3 | 4 | 5 | 6 | 7 | Avg. |
|---|---|---|---|---|---|---|
| **Qwen2.5-3B-Instruct** | 0.09 | 0.10 | 0.03 | 0.05 | 0.02 | 0.06 |
| + DAPO | 0.68 | 0.61 | 0.45 | 0.42 | 0.31 | 0.49 |
| **+ DAPO + Reweight** | 0.72 | 0.68 | 0.52 | 0.51 | 0.47 | 0.58 (↑17.4%) |
| **+ DAPO + Lopti** | 0.77 | 0.73 | 0.60 | 0.58 | 0.50 | 0.64 (↑28.7%) |
| **+ DAPO + Reweight + Lopti** | 0.85 | 0.88 | 0.71 | 0.70 | 0.58 | 0.74 (↑50.6%) |
| **Qwen2.5-7B-Instruct-1M** | 0.22 | 0.15 | 0.08 | 0.10 | 0.02 | 0.11 |
| + DAPO | 0.96 | 0.93 | 0.81 | 0.76 | 0.67 | 0.82 |
| **+ DAPO + Reweight** | 0.97 | 0.97 | 0.92 | 0.87 | 0.86 | 0.92 (↑11.1%) |
| **+ DAPO + Lopti** | 0.99 | 0.99 | 0.97 | 0.92 | 0.90 | 0.95 (↑15.5%) |
| **+ DAPO + Reweight + Lopti** | 1.00 | 0.99 | 0.98 | 0.96 | 0.91 | 0.97 (↑17.2%) |

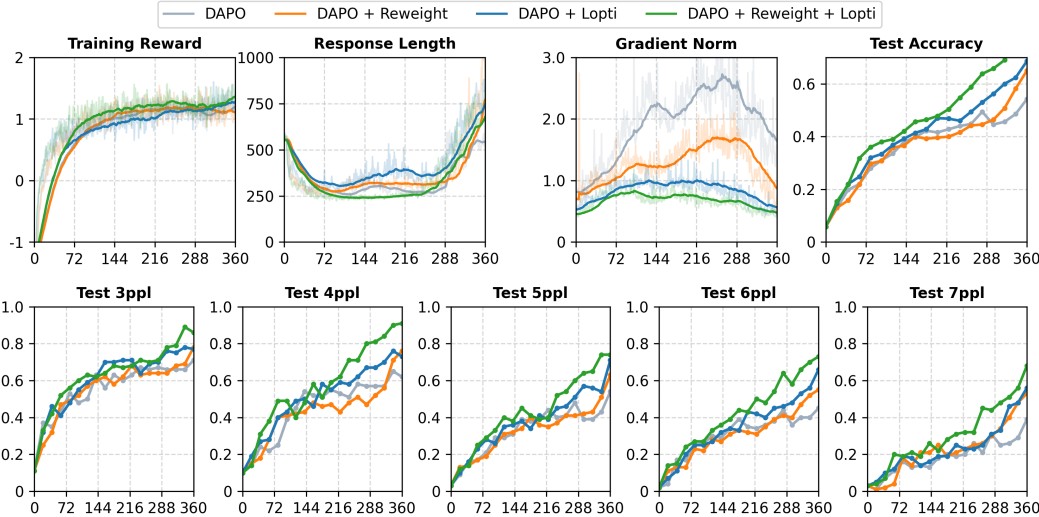

Figure 14: Experimental records of `Qwen2.5-3B-Instruct` trained with DAPO on the K&K Logic Puzzle dataset.

DAPO utilizes the same training objective as defined in Eq. 1. In our experiments, we validate DAPO on the K&K Logic Puzzle dataset using two base models: `Qwen2.5-3B-Instruct` and `Qwen2.5-7B-Instruct-1M`. All hyperparameters of DAPO are kept consistent with those used for GRPO, as described in Appendix B. It is important to note that during the sampling process, DAPO filters out prompts that yield homogeneous responses. This filtering operation results in variability in the total number of training steps across different experimental settings, as the filtered prompts differ in each training run. To ensure a fair comparison, we do not fix the number of training epochs at 5; instead, we fix the total number of training steps. For the 3B model, we set the training steps to 360, consistent with the experiments conducted for GRPO and REINFORCE++. However, for the 7B model, DAPO converges after 300 steps, and the prevalence of homogeneous responses during the sampling process triggers early stopping. Consequently, we set the training steps to 300 for the 7B model. The experimental results are presented in Table 8 and Figures 14 15.

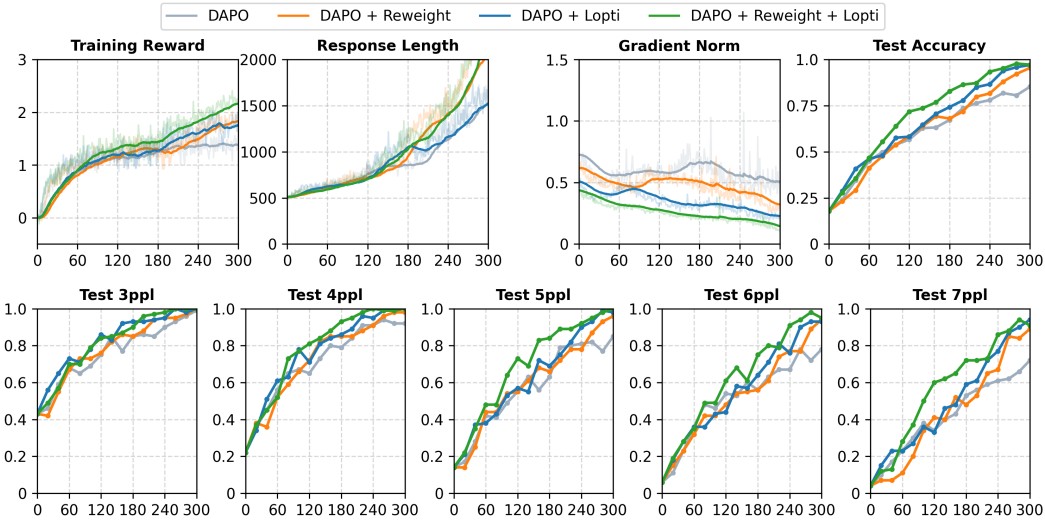

Figure 15: Experimental records of `Qwen2.5-7B-Instruct-1M` trained with DAPO on the K&K Logic Puzzle dataset.

## D.2 ADDITIONAL LARGE LANGUAGE MODELS

To demonstrate the generalization capability of our proposed method beyond Qwen-series models, we further evaluate its performance on LLaMA-series (Dubey et al., 2024) and DeepSeek-series (Guo et al., 2025) models in Appendices D.2.1 and D.2.2, respectively.

### D.2.1 LLAMA

We adopt two models from the LLaMA-series (Dubey et al., 2024): `LLaMA-3.2-3B-Instruct` and `LLaMA-3.1-8B-Instruct` for our experiments. We evaluate our methods with GRPO on the K&K Logic Puzzle dataset. All other hyperparameters are kept consistent with those in Appendix B. The experimental results are presented in Table 9, Figure 16, and Figure 17. These results closely resemble those obtained with Qwen-series models, as shown in Table 4, Figure 7, and Figure 8.

Table 9: Experimental records of `LLaMA-3.2-3B-Instruct` & `LLaMA-3.1-8B-Instruct` trained with GRPO on the K&K Logic Puzzles dataset. For *Advantage Reweight*, $\alpha = 0.3$, and for *Lopti*, $\eta = 0.5$.

| Model | Difficulty by Number of People | | | | | |
|---|---|---|---|---|---|---|
| | 3 | 4 | 5 | 6 | 7 | Avg. |
| **LLaMA-3.2-3B-Instruct** | 0.00 | 0.00 | 0.01 | 0.00 | 0.01 | 0.00 |
| GRPO | 0.47 | 0.33 | 0.29 | 0.27 | 0.13 | 0.30 |
| **GRPO + Reweight** | 0.58 | 0.51 | 0.33 | 0.36 | 0.23 | 0.40 (↑33.3%) |
| **GRPO + Lopti** | 0.73 | 0.70 | 0.59 | 0.49 | 0.43 | 0.59 (↑96.7%) |
| **GRPO + Reweight + Lopti** | 0.73 | 0.76 | 0.58 | 0.55 | 0.45 | 0.62 (↑106.7%) |
| **LLaMA-3.1-8B-Instruct** | 0.05 | 0.01 | 0.03 | 0.00 | 0.00 | 0.02 |
| GRPO | 0.86 | 0.88 | 0.77 | 0.72 | 0.68 | 0.78 |
| **GRPO + Reweight** | 0.89 | 0.92 | 0.86 | 0.81 | 0.78 | 0.85 (↑9.0%) |
| **GRPO + Lopti** | 0.90 | 0.95 | 0.89 | 0.87 | 0.84 | 0.88 (↑12.8%) |
| **GRPO + Reweight + Lopti** | 0.94 | 0.97 | 0.89 | 0.86 | 0.82 | 0.90 (↑15.4%) |

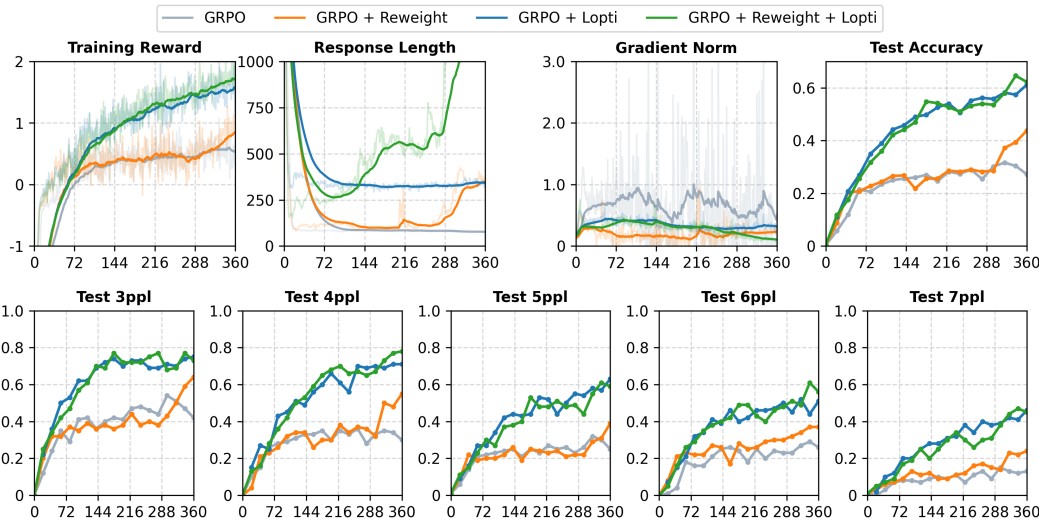

Figure 16: Experimental records of `LLaMA-3.2-3B-Instruct` trained with GRPO on the K&K Logic Puzzle dataset.

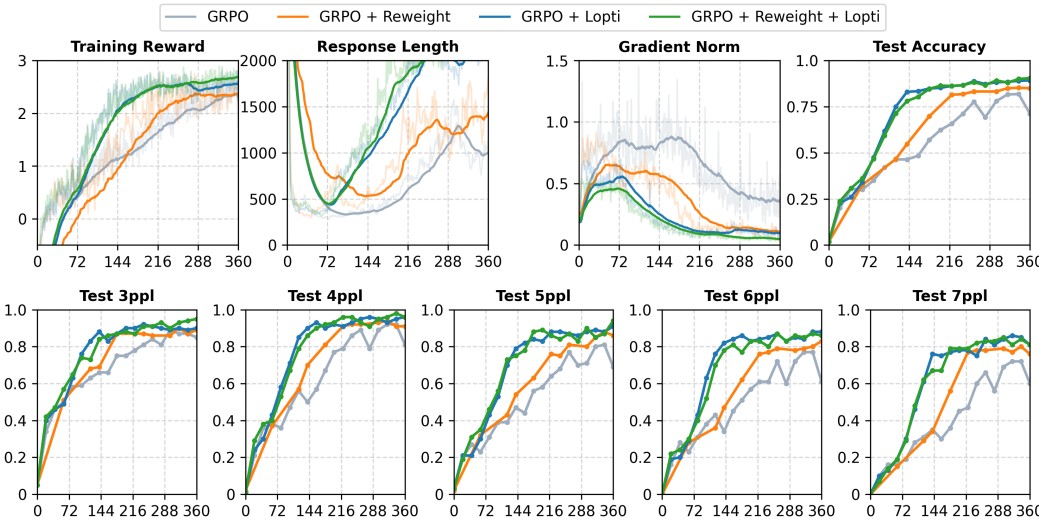

Figure 17: Experimental records of `LLaMA-3.1-8B-Instruct` trained with GRPO on the K&K Logic Puzzle dataset.

### D.2.2 DeepSeek-R1-Distill

As discussed in Section 5.2 and illustrated in Figure 10, the test accuracy curve on the math-related dataset converges to a specific value within 100 steps and subsequently exhibits only minor fluctuations. Consequently, the improvements introduced by *Advantage Reweighting* and *Lopti* are not significant. This phenomenon is common for base models with limited reasoning capability, as they encounter similar data during the pre-training phase and their capability is constrained by the number of model parameters.

To observe the continuous learning behavior on the math-related dataset and reveal the capability of our proposed methods, we conduct experiments on the ORZ dataset with `DeepSeek-R1-Distill-1.5B`. All hyperparameters are set the same as those in Appendix B.

The experimental results are recorded in Table 10 and Figure 18. It is evident that the gap between the baseline GRPO and GRPO enhanced with *Advantage Reweighting* or *Lopti* becomes increasingly larger for both the reward curve and the test curve. Compared with the experimental results shown in Figure 10, the improvements brought by our proposed methods are more pronounced.

Table 10: Experimental results on of `DeepSeek-R1-Distill-1.5B` trained with GRPO on ORZ dataset.

| Dataset | Algorithms | Olympiad Bench | Minerva | MATH 500 | AMC avg@16 | AIME24 pass@16 | AIME24 avg@16 | Avg. all |
|---------|------------|----------------|---------|----------|------------|----------------|---------------|----------|
| | **DeepSeek-R1-Distill-1.5B** | 24.81 | 9.93 | 55.40 | 34.64 | 23.33 | 9.58 | 26.28 |
| **ORZ** | + GRPO | 35.84 | 19.51 | 73.90 | 49.85 | 33.26 | 14.78 | 37.86 |
| | + GRPO + Reweight | 40.02 | 19.73 | 77.20 | 53.97 | 32.22 | 17.50 | 40.11 |
| | + GRPO + Lopti | 37.25 | 20.34 | 75.67 | 51.66 | 34.44 | 14.72 | 39.01 |

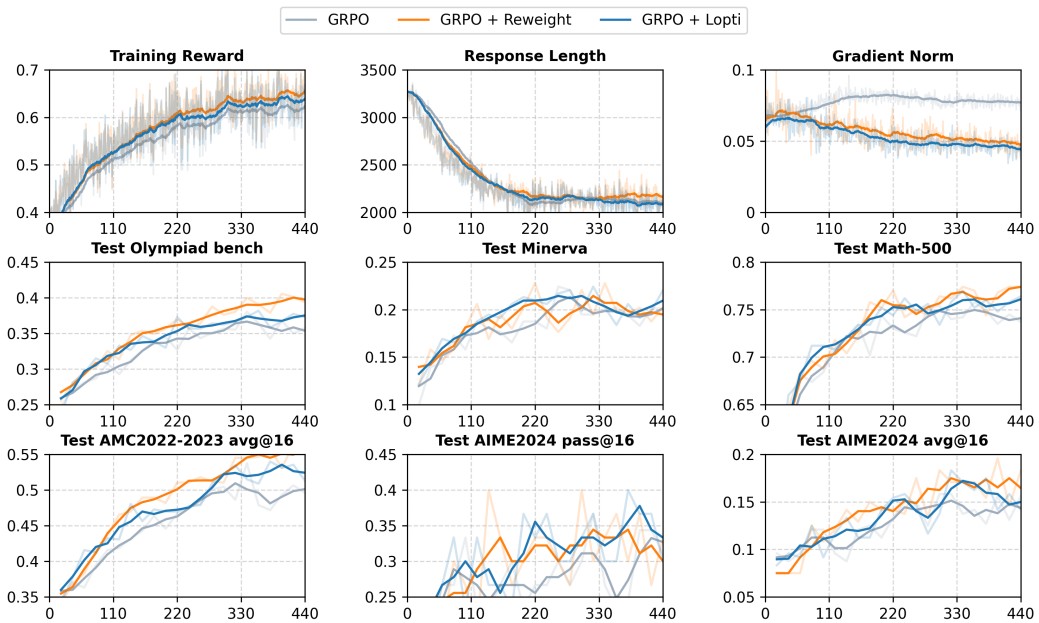

Figure 18: Experimental records of `DeepSeek-R1-Distill-1.5B` trained with GRPO on the ORZ-math dataset.

# E ON THE CONTRADICTORY CONCLUSIONS OF THE 80/20 RULE

It is noteworthy that the conclusions from our paper are somewhat contradictory to those of the 80/20 Rule of RL introduced by Wang et al. (2025). Specifically, in this work, we emphasize that high-probability tokens are equally important as low-probability ones, and that the over-dominant effect of low-probability tokens may impede the learning of high-probability tokens. In contrast, Wang et al. (2025) argue that only high-entropy tokens (which are somewhat equivalent to low-probability tokens) matter, and that all low-entropy tokens (high-probability ones) can be completely dropped during the training process.

We would like to emphasize that the fundamental reason lies in the hyperparameter settings used in the experiments. We follow the settings from Logic-RL (Xie et al., 2025), while Wang et al. (2025) follow those from DAPO (Yu et al., 2025). In our setting, we set the update batch size to half of the total samples per RL step, i.e., the sampled trajectories are separated into $n = 2$ batches for updating. This configuration was common before RL scaled up to models of extremely large size. For extremely large models, switching between inference and training engines can be extremely slow due to heavy I/O costs. For this reason, DAPO samples a large batch and updates it separately across $n = 16$

gradient steps. This means that only the first batch is on-policy, while the rest are off-policy, causing the clipping mechanism to be activated. It is observed that in such off-policy cases, low-probability tokens are clipped to a large extent, leading to an overemphasis on high-probability tokens, which are less likely to be clipped in the later batches of updates. This phenomenon can be mitigated by the 80/20 rule. However, for relatively on-policy training (when $n$ is small), the 80/20 rule does not work. To verify this analysis, we conduct experiments as shown in Figure 19. It is noteworthy that the 80/20 rule degrades the performance of the baseline DAPO for $n \leq 8$ and only improves performance when $n = 16$.

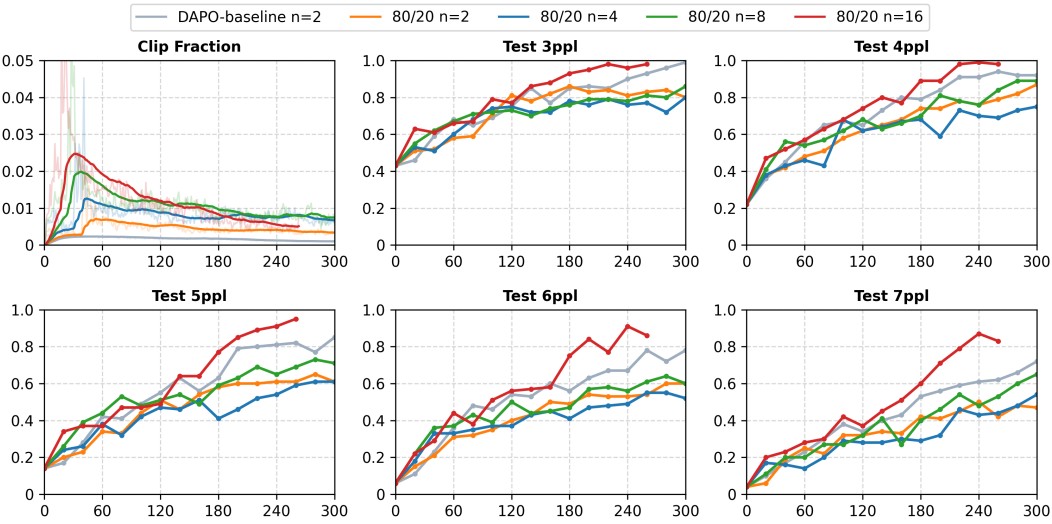

Figure 19: The effect of the 80/20 rule (Wang et al., 2025) with different numbers of sampled batch shards on K&K Logic Puzzle dataset.

## F LIMITATIONS

One limitation of our study lies in the additional computational overhead introduced by *Lopti*. As detailed in Appendix C.2, the updating process requires twice the amount of time as it splits the tokens in a batch into two groups and performs updates twice. However, we also propose an alternative method, *Advantage Reweighting*, which incurs negligible computational cost while achieving even greater improvements on the math-related dataset compared to *Lopti*.

