# OpenReview forum: "Do Not Let Low-Probability Tokens Over-Dominate in RL for LLMs"
_ICLR.cc/2026/Conference — ICLR 2026 Poster_

### Official Review · Reviewer_xEgW · 2025-10-28

**Soundness:** 4
**Presentation:** 3
**Contribution:** 3
**Rating:** 6
**Confidence:** 4

**Summary:**

This work first identifies the pattern that tokens with low probability could induce significantly higher gradients than those with high probability tokens. Basing on this observation, this work then theoretically prove that the gradient norms are proportional to (1-$\pi$), where $\pi$ is the token probability.  Then, this work proposes two algorithms, advantage reweighing and low-probability token isolation to alleviate the issue.

The proposed method are evaluated on two kinds of tasks: K&K logic puzzles and mathematical reasoning. On both tasks, the proposed methods can outperform the original GRPO. Furthermore, on K&K, the results show that further improvement can be obtained by combining advantage reweighing and low-probability token isolation.

**Strengths:**

1, This work rigorously describes how the gradient norm is decided by token probability, during online RL training. This provides valuable insights to the community, and encourages exploration on token-level reward design.

2, The two proposed methods are sound, and straightforward. According to empirical results, they can effectively improve performance over the original GRPO.

3, I especially appreciate the in-depth ablation studies. They demonstrate: 1, how different tokens with different probs affect the training dynamics; 2, different hyper-parameters affect the training dynamics compared to GRPO.

**Weaknesses:**

1, The first concern is with the hyper parameters. According to Figure 6, the choices of $\alpha$ and $\eta$ significantly affect the training dynamics. But considering that the training and test curves are similar, introducing a validation mechanism would be very helpful. Would you like to discuss such mechanism?

2, On a philosophical level, I am not sure if such fixing (improving the effect of high-probability grad norms) is necessary, especially on positive samples. Specifically, those tokens are already of high probability so that improving their grad norm can only marginally change their distribution. Figure (6) is helpful in demonstrating the effects of low and high probability tokens. But those experiments are different from gradient update. Specifically, inclusions of different probability tokens in figure (6) is hard: tokens are either updated or not updated; but in gradient update, all tokens are updated, but the update magnitudes (gradient norm) are different.

3, I do appreciate the bounds over the gradient norm. But I am also concerned that $c_l$ and d_l$ could be sufficiently large, dominating the effect of token probability.

**Questions:**

1, Could you also report the curves on mathematical tasks (e.g., MATH 500)?

2, How did you select the hyper parameters in Table 3?

3, Do you have any hypothesis for whether combining advantage reweighing and Lopti did not improve for math problems?

---

> ### Author Response · Authors · 2025-11-19
> **Response Part 1 of 2**
>
> We thank Reviewer xEgW for providing the positive feedback and recognizing the importance of our work. Please see our response to your concerns below.
>
> ### Weaknesses
>
> 1. The first concern is with the hyper parameters. According to Figure 6, the choices of $\alpha$ and $\eta$ significantly affect the training dynamics. But considering that the training and test curves are similar, introducing a validation mechanism would be very helpful. Would you like to discuss such mechanism?
>
> In our observations, the optimal hyperparameter settings for our proposed methods are related to task specification rather than to the base model utilized.
>
> We observed an interesting phenomenon: for reasoning tasks where the model uses natural language for inference, low-probability tokens occur more frequently in the sampled batch, and consequently, their dominant effect is more pronounced. Therefore, integrating *Advantage Reweighting* and *Lopti* yields better results. Conversely, for tasks such as mathematics, where the model employs specialized mathematical notation for inference, low-probability tokens occur less frequently, and the dominant effect is less pronounced under such circumstances. For more details, please refer to our response to Question 3 and Appendix C.3 (Page 22-23) of the revised version of our paper.
>
> Regarding the hyperparameter $\alpha$ for *Advantage Reweighting*: for tasks where low-probability tokens exhibit a greater dominant effect (such as reasoning tasks), it should be set to $0.2$--$0.3$ to achieve optimal performance. For tasks where the dominant effect is weaker (such as math tasks), it should be set to $0.1$ for optimal performance. As for the hyperparameter $\eta$ for *Lopti*, it is robust across tasks, and setting it to $0.5$ is sufficient for all tasks. We have added the relevant description in the revised version of our paper (Lines 891-901, Page 17).
>
> 2. On a philosophical level, I am not sure if such fixing (improving the effect of high-probability grad norms) is necessary, especially on positive samples. Specifically, those tokens are already of high probability so that improving their grad norm can only marginally change their distribution. Figure (6) is helpful in demonstrating the effects of low and high probability tokens. But those experiments are different from gradient update. Specifically, inclusions of different probability tokens in figure (6) is hard: tokens are either updated or not updated; but in gradient update, all tokens are updated, but the update magnitudes (gradient norm) are different.
>
> Indeed, improving only the effect of positive high-probability tokens may not seem intuitively necessary. However, please note that our methods are not restricted to improving the effect of high-probability tokens (with probability greater than 0.75). Rather, **they balance the updates across all tokens and can increase the correct update ratio of all possitive tokens with probability greater than 0.25, as presented in Figure 3 of our main paper**. We believe such an effect is necessary and can introduce significant gains during the RL training process.
>
> We agree with your point that the training curves presented in Figure 6(a) provide limited information from the gradient perspective. However, the primary purpose of this ablation study is to underscore the indispensability of high-probability tokens during the RL training process. As evidenced by the results, masking these tokens leads to a significant degradation in the performance of the baseline GRPO, confirming their crucial role.
>
>
> 3. I do appreciate the bounds over the gradient norm. But I am also concerned that $c_l\leq\sigma_\min(J_f)$ and $d_l\geq\sigma_\max(J_f)$ could be sufficiently large, dominating the effect of token probability.
>
> Thanks for the insightful question. Owing to the use of ResNet-style residual connections in the Transformer decoder block, the analysis can be carried out as follows.
>
> We first define the residual map
> $$f(a)=a+h(a),$$
> where $h$ denotes the residual function, and $f$ corresponds to either the attention block or the feedforward block in a Transformer decoder. We further assume that $h$ is globally $L$-Lipschitz with $L\leq 1$. By linearity of the derivative, we have
> $$\frac{\partial f(a)}{\partial a} = \frac{\partial a}{\partial a} + \frac{\partial h(a)}{\partial a}$$
> $$J_f(a) = I + J_h(a),$$
> where $J_f$ and $J_h$ denote the Jacobian matrices of $f$ and $h$, respectively.
>
> Let $||\cdot||$ denote the spectral norm. By the triangle inequality and the fact that the spectral norm equals the largest singular value, we obtain
> $$||J_f(a)|| = ||I + J_h(a)||$$
> $$||J_f(a)|| \leq ||I|| + ||J_h(a)||$$
> $$\sigma_\max(J_f(a)) \leq 1 + \sigma_\max(J_h(a))$$
> $$\sigma_\max(J_f(a)) \leq 1 + L.$$
>
> (Due to the space limit, please refer to Part 2 for the rest derivation.)

---

> > ### Author Response · Authors · 2025-11-19
> > **Response Part 2 of 2**
> >
> > In the other direction, we consider the smallest singular value of $J_f(a)$. By definition,
> > $$\sigma_\min(J_f(a)) = \min_{||v||=1} ||J_f(a)\cdot v||= \min_{||v||=1} ||v + J_h(a)\cdot v||.$$
> >
> > Using the reverse triangle inequality for vectors, we have
> > $$||v + J_h(a)\cdot v|| \geq \vert ||v|| - ||J_h(a)\cdot v||\vert = \vert 1 - ||J_h(a)\cdot v||\vert$$
> > for any unit vector $||v||=1$. Moreover, by the definition of the spectral norm,
> > $$||J_h(a)\cdot v||\leq||J_h(a)||=\sigma_\max(J_h(a))\leq L$$
> > Therefore,
> > $$\sigma_\min(J_f(a)) \geq 1-L.$$
> > Combining the bounds on the largest and smallest singular values, we obtain
> > $$1-L \leq \sigma_\min(J_f(a)) \leq \sigma_\max(J_f(a)) \leq 1+L,$$
> > which means that all singular values of $J_f(a)$ are tightly concentrated around 1 when $L$ is sufficiently small.
> >
> > To verify the above theoretical derivation, we further record the singular values of the Jacobians of the attention and feedforward blocks in the input decoder layer (layer index 0) of *Qwen2.5-7B-Instruct-1M*. The statistics are computed over 300 tokens from a single sentence and are summarized in the following table.
> >
> > |Block|Indicator|Mean|Std|Max|Min|
> > |-|-|-|-|-|-|
> > |Attention|$\sigma_\max(J_f(a))$|1.476|0.097|2.614|1.412|
> > |-|$\sigma_\min(J_f(a))$|0.673|0.037|0.704|0.343|
> > |Feedforward|$\sigma_\max(J_f(a))$|1.105|0.041|1.247|1.059|
> > |-|$\sigma_\min(J_f(a))$|0.917|0.022|0.948|0.847|
> >
> > In conclusion, $c_l$ and $d_l$ cannot be extremely large, as they are all close to 1 for a residual connected block.
> >
> > ### Questions
> >
> > 1. Could you also report the curves on mathematical tasks (e.g., MATH 500)?
> >
> > Due to space constraints, we initially presented all training curves in Appendix D. For mathematics-related tasks, we report training reward, response length, gradient norm, and all evaluation metrics on the test sets (MATH-500, OlympiadBench, MINERVA Math, AMC 2022-2023, and AIME 2024). These figures can be found in Appendix C.1.2 (Page 20-22) of the revised manuscript.
> >
> > 2. How did you select the hyper parameters in Table 3?
> >
> > Please refer to our response to your listed Weakness 1.
> >
> > 3. Do you have any hypothesis for whether combining advantage reweighing and Lopti did not improve for math problems?
> >
> > We indeed have some interesting findings regarding such phenomenon. We believe that the root of this problem lies in the characteristics of the specific nature of the task at hand.
> >
> > - In our observations, when the task involves "logic puzzle", LLMs encounter a greater number of "high-entropy" positions during the auto-regressive generation process. This remains true even when we lower the temperature to 0.7 (as opposed to the typical setting of 1.0 used in mathematical tasks). As a result, the proportion of "low-probability tokens" in the sampled batch increases. In such scenarios, the dominance effect of low-probability tokens becomes more pronounced, allowing both "Advantage Reweighting" and "Lopti" to perform effectively. When used independently, there is no significant difference in performance; when combined, they can even yield superior results.
> >
> > - In contrast, for math-related tasks, high-entropy positions are encountered less frequently during the generation process, leading to a lower proportion of low-probability tokens in the sampled batch compared to logic puzzles. This may be attributed to the greater textual inertia and determinism inherent in mathematical language. Consequently, combining the two methods in this context can excessively diminish the influence of low-probability tokens in the gradient updates, ultimately resulting in suboptimal performance.
> >
> > We hope this clarification addresses your concerns. We have provided a more detailed analysis and discussion of these points in Appendix C.3 (Page 22-23) of the revised version of our paper.

---

> > > ### Comment · Reviewer_xEgW · 2025-11-26
> > > **Thanks for the response.**
> > >
> > > First, I thank the authors for their informative responses. My concerns have been addressed. So, I will raise my score. I believe this work makes valuable contributions to the community.
> > >
> > > Just one minor question, in the rebuttal, you assume the Lipschitz to be no greater than 1. Is this necessary? As I do not think this assumption is used in the later derivations.

---

> > > > ### Author Response · Authors · 2025-11-26
> > > > **Thanks for raising score**
> > > >
> > > > We sincerely thank Reviewer xEgW for acknowledging our rebuttal and for the increased score.
> > > >
> > > > Regarding your final question, we agree that the assumption of the Lipschitz constant being no greater than 1 is not necessary. Without this assumption, the lower bound of $\sigma_\min(J_f(a))$ becomes $\sigma_\min(J_f(a)) \geq |1-L|$ according to the reverse triangle inequality. This modification does not affect the core conclusion we would like to convey.

---

### Official Review · Reviewer_jZRG · 2025-10-30

**Soundness:** 2
**Presentation:** 3
**Contribution:** 2
**Rating:** 2
**Confidence:** 4

**Summary:**

This research identifies a critical but previously overlooked inefficiency in the reinforcement learning (RL) process for Large Language Models (LLMs): the disproportionate dominance of low-probability tokens in policy gradient updates. The paper argues that this phenomenon, termed the "tyranny of the unlikely token," actively hinders the model's ability to refine its core knowledge. To address this, the paper proposes the advantage reweighting and Low-probability Token Isolation (Lopti). Experiments on complex reasoning and math tasks using GRPO demonstrate that both methods significantly outperform the baseline.

**Strengths:**

1. The research is built on a solid theoretical foundation, with Proposition 4.2 mathematically demonstrating that a token's gradient norm is inversely related to its probability. This elevates empirical observation to a predictable phenomenon.
2. The paper excels at communicating a complex technical subject. The narrative is logical, and the framing of the issue as the "tyranny of the unlikely token" is both memorable and effective.

**Weaknesses:**

1. The author claim that low-probability tokens dominate model updates during RL training and that this dominance may impede the precise adjustment of the probability distribution across all tokens.   In fact, the high-entropy minority tokens drive effective reinforcement learning for llm reasoning, we do not need to adjust the probability distribution across all tokens based on RL.

2. RL Algorithm Specificity: Experiments were conducted exclusively with GRPO. All experiments used models from the Qwen2.5 family.  There is no other strong comparable baselines on the benchmark.

3. Acknowledging that Lopti doubles update time is treated as a minor caveat. A 100% increase in backward pass cost is prohibitive for most large-scale training operations, rendering Lopti impractical for all but the most specialized applications.

4.The paper offers no guidance on how to co-tune the two new hyperparameters (α and η).




[1] Beyond the 80/20 rule: High-entropy minority tokens drive effective reinforcement learning for llm reasoning, arXiv 2025.

**Questions:**

The solutions are presented without comparison to existing techniques for managing noisy gradients.

---

> ### Author Response · Authors · 2025-11-19
> **Response Part 1 of 2**
>
> We thank Reviewer jZRG for the valuable feedback. We would like to address your concerns point by point below.
>
> ### Weaknesses
> 1. The author claim that low-probability tokens dominate model updates during RL training and that this dominance may impede the precise adjustment of the probability distribution across all tokens. In fact, the high-entropy minority tokens drive effective reinforcement learning for llm reasoning, we do not need to adjust the probability distribution across all tokens based on RL.
>
> Thank you for introducing the relevant work. We find that the conclusions from our paper are somewhat contradictory to those of the 80/20 Rule of RL introduced by [1]. Specifically, in this work, we emphasize that high-probability tokens are equally important as low-probability ones, and that the over-dominant effect of low-probability tokens may impede the learning of high-probability tokens. In contrast, [1] argues that only high-entropy tokens (which are somewhat equivalent to low-probability tokens) matter, and that all low-entropy tokens (high-probability ones) can be completely dropped during the training process.
>
> First, we would like to clarify that the 80/20 rule does not apply to all scenarios. As shown in Figure 6(a) (Page 9) of our main text, when we mask those high-probability tokens, no gains are observable, while the degradation in performance is significant.
>
> Second, we would like to emphasize that the fundamental reason lies in the hyperparameter settings used in the experiments. We follow the settings from Logic-RL, while [1] follows those from DAPO [2]. In our setting, we set the update batch size to half of the total samples per RL step, i.e., the sampled trajectories are separated into $n=2$ batches for updating. This configuration was common before RL scaled up to models of extremely large size. For extremely large models, switching between inference and training engines can be extremely slow due to heavy I/O costs. For this reason, DAPO samples a large batch and updates it separately across $n=16$ gradient steps. This means that only the first batch is on-policy, while the rest are off-policy, causing the clipping mechanism to be activated.
> It is observed that in such off-policy cases, low-probability tokens are clipped to a large extent, leading to an overemphasis on high-probability tokens, which are less likely to be clipped in the later batches of updates. This phenomenon can be mitigated by the 80/20 rule. However, for relatively on-policy training (when $n$ is small), the 80/20 rule does not work. To verify this analysis, we conduct experiments in Appendix E of the revised manuscript (Page 29-30). It is noteworthy that the 80/20 rule degrades the performance of the baseline DAPO for $n\leq 8$ and only improves performance when $n=16$.
>
> We would like to emphasize that online-RL algorithms inherently favor on-policy updates, whereas off-policy updates yield suboptimal performance. Accordingly, large values of $n$ are most appropriate for training large-scale models (≥32B parameters), where I/O overhead becomes a dominant constraint. Conversely, for smaller-scale models, it is more usual to employ smaller $n$ values, which aligns well with the design principles of our methods.
>
> 2. RL Algorithm Specificity: Experiments were conducted exclusively with GRPO. All experiments used models from the Qwen2.5 family. There is no other strong comparable baselines on the benchmark.
>
> As discussed in Section 5, our proposed methods are not restricted to GRPO and demonstrate significant potential across all policy-gradient-based RL algorithms. They can be regarded as an orthogonal enhancement applicable to various approaches. In the original version of our paper, we provided additional experiments on REINFORCE++, which can be found on Page 23-25 of the current version.
>
> Furthermore, to demonstrate the broader generalization of our proposed approach, we have included additional experimental results on several models, including LLaMA-3.1-8B-Instruct, LLaMA-3.2-3B-Instruct, and DeepSeek-R1-Distill-1.5B, and have incorporated a new baseline algorithm, DAPO. Please see our global response for details.
>
> We believe these additional experimental results adequately address your concerns regarding the generalizability and robustness of our approach.

---

> > ### Author Response · Authors · 2025-11-19
> > **Response Part 2 of 2**
> >
> > 3. Acknowledging that Lopti doubles update time is treated as a minor caveat. A 100% increase in backward pass cost is prohibitive for most large-scale training operations, rendering Lopti impractical for all but the most specialized applications.
> >
> > Please note that, in online RL4LLM algorithms, the most time-consuming component is the sampling (rollout) phase rather than the parameter update step. This is because sampling requires auto-regressive generation, whereas the updates can be computed in parallel. This effect is particularly pronounced for reasoning tasks with extremely long reasoning trajectories. We report the detailed computational cost in Table 6 of the main paper (Page 22). The *Lopti* operation increases the total training time by approximately 30–40%. We also explicitly list this issue as a limitation of our proposed methods in Appendix E. Furthermore, our proposed alternative approach, *Advantage Reweighting*, does not suffer from this overhead.
> >
> > Moreover, we would like to point out that *DAPO*[2] incurs roughly three times the computational cost in the sampling phase, yet this has not prevented it from becoming a widely adopted and well-accepted solution. In the following table, we present a comparison of the computational overhead incurred by DAPO and our proposed Lopti relative to the baseline GRPO.All experiments are conducted on a single machine equipped with eight NVIDIA H20 96GB GPUs, using Qwen2.5-7B-Instruct-1M as the base model and the K&K Logic Puzzle dataset.
> >
> > | Procedure | GRPO | DAPO | GRPO+Lopti|
> > |-|-|-|-|
> > |Training | 24.9 | 24.1 | 47.5 |
> > |Sampling | 31.1 | 71.6 | 28.6 |
> > | Others | 6.6 | 7.2 | 6.1 |
> > |Total | 62.6 | 102.9 | 82.2 |
> >
> >
> >
> > 4. The paper offers no guidance on how to co-tune the two new hyperparameters (α and η).
> >
> > In our observations, the optimal hyperparameter settings for our proposed methods are related to task specification rather than to the base model utilized.
> >
> > We observed an interesting phenomenon: for reasoning tasks where the model uses natural language for inference, low-probability tokens occur more frequently in the sampled batch, and consequently, their dominance effect is more pronounced. Therefore, integrating *Advantage Reweighting* and *Lopti* yields better results. Conversely, for tasks such as mathematics, where the model employs specialized mathematical notation for inference, low-probability tokens occur less frequently, and the dominance effect is less pronounced under such circumstances. For more details, please refer to Appendix C.3 of the revised version of our paper.
> >
> > Regarding the hyperparameter $\alpha$ for *Advantage Reweighting*: for tasks where low-probability tokens exhibit a greater dominant effect (such as reasoning tasks), it should be set to $0.2$--$0.3$ to achieve optimal performance. For tasks where the dominant effect is weaker (such as math tasks), it should be set to $0.1$ for optimal performance. As for the hyperparameter $\eta$ for *Lopti*, it is robust across tasks, and setting it to $0.5$ is sufficient for all tasks. When combining both techniques (for reasoning tasks), maintaining these optimal settings yields good performance. We have added the relevant description in the revised version of our paper (Lines 891-901, Page 17).
> >
> > ### Questions
> > 1. The solutions are presented without comparison to existing techniques for managing noisy gradients.
> >
> > We would like to note that our methods are orthogonal to policy-gradient-based RL algorithms and have great potential to be integrated into existing methods. We hope the added experiments address your concerns.
> >
> > [1]. Want et al. Beyond the 80/20 Rule: High-Entropy Minority Tokens Drive Effective Reinforcement Learning for LLM Reasoning.
> >
> > [2]. Yu et al. DAPO: An Open-Source LLM Reinforcement Learning System at Scale.

---

### Official Review · Reviewer_5xRJ · 2025-10-31

**Soundness:** 4
**Presentation:** 4
**Contribution:** 3
**Rating:** 8
**Confidence:** 3

**Summary:**

The paper identifies and analyzes an overlooked source of update bias in RL training of LLMs: low-probability tokens produce larger gradient magnitudes and can therefore dominate parameter updates. The authors (1) derive bounds showing token-wise gradient norms scale with $1−\pi$, (2) propose two mitigation methods: Advantage Reweighting and Low-Probability Token Isolation (Lopti), and (3) demonstrate consistent empirical gains on several benchmarks.

**Strengths:**

1. The theoretical derivation and supporting experiments are well aligned; the motivation is lucid and convincing.
2. The proposed methods are low-cost, effective, and easy to implement in practice.
3.  The authors provide careful experiments and analysis, including evaluations on multiple algorithms, multiple domains, and several ablations.

**Weaknesses:**

See questions below.

**Questions:**

1. The paper evaluates only naive GRPO and REINFORCE++ baselines. It’s better to consider whether widely used GRPO variants (for example, DAPO) might interact with or mitigate the same gradient-magnitude imbalance.
2. To demonstrate broader generalization, It’s better to add experiments across different model families and sizes (for example, Qwen-3 and Llama-3.1).

---

> ### Author Response · Authors · 2025-11-19
>
> We thank Reviewer 5xRJ for providing the positive feedback and recognizing the importance of our work. Please see our response to your questions below.
>
> ### Questions
> 1. The paper evaluates only naive GRPO and REINFORCE++ baselines. It’s better to consider whether widely used GRPO variants (for example, DAPO) might interact with or mitigate the same gradient-magnitude imbalance.
>
> Thanks for your valuable suggestion. We do agree that intergrate DAPO will further prove the generalization capability of our proposed methods. We have supplemented the experimental results on DAPO as in Table 8, Figures 14-15, Page 25-27. For a quick overview of the results, please refer to the global response Point 3 for more details.
>
>
> 2. To demonstrate broader generalization, It’s better to add experiments across different model families and sizes (for example, Qwen-3 and Llama-3.1).
>
> Following your suggestion, we conduct experiments on LLaMA-series models (*LLaMA-3.1-8B-Instruct* and *LLaMA-3.2-3B-Instruct*). The detailed results can be found in Table 9, Figures 16-17, Page 27-28 of the updated version of our paper. For a quick overview, please refer to the global response Point 4.

---

> > ### Comment · Reviewer_5xRJ · 2025-11-27
> > **Reply to Authors**
> >
> > I thank the authors for the additional results. Most of my concerns have been addressed, and I will maintain my positive rating and raise the confidence score.

---

### Official Review · Reviewer_A6Dz · 2025-10-31

**Soundness:** 3
**Presentation:** 3
**Contribution:** 3
**Rating:** 6
**Confidence:** 3

**Summary:**

This paper claims that low-probability tokens have higher gradient norms which over-dominates the gradient update when doing RLVR to an extent that high-probability tokens might update even in the opposite direction. After showing this empirically, the show via mitigating this via isolating or re-weighting the low-probability tokens, we have a better RLVR algorithm.

**Strengths:**

I think this paper's main claim is new to the field: the gradient of the low probability tokens are dictating the changes in the probability of high probability tokens (not the advantages of those high-probability tokens) because their gradient magnitude is large. The paper shows mitigating this improves results on K&K puzzle. I am not familiar with this puzzle, but it seems it is a challenging task. I think the K&K results are the strongest in supporting the evidence.

**Weaknesses:**

The paper points out a very interesting observation. However, the math results are not consistent with the theory: their methods and GRPO are almost achieving the same score. I think it is because they are testing this on a R1-Zero style scenario where they start from a base model. Papers that do RL on base models show this quick recovery of some performance the curves are flat afterwards. I think the paper should have been done on a native reasoning model on R1-Distill-1.5B as they show usually continued progress even in math problems. Therefore, I would say the experiments are not providing the  significant signal that this algorithm scales. In terms of the reasoning to support the conclusion that mitigating the dominance of low-probability tokens is key in RLVR, while the K&K results support the claim, we don't know if this holds in more serious scenarios like math problems. I don't know if K&K is considered a task that its performance forecasts the performance on downstream hard reasoning.

**Questions:**

1-Is there evidence that K&K is a hard reasoning benchmark?
2-Have you tried this method on native reasoning models, not base models, and observe improvement?

---

> ### Author Response · Authors · 2025-11-19
>
> We sincerely acknowledge Reviewer A6Dz for providing the insightful feedback. Please see our response to your concerns below.
>
> ### Weaknesses
>
> 1. The paper points out a very interesting observation. However, the math results are not consistent with the theory: their methods and GRPO are almost achieving the same score. I think it is because they are testing this on a R1-Zero style scenario where they start from a base model. Papers that do RL on base models show this quick recovery of some performance the curves are flat afterwards. I think the paper should have been done on a native reasoning model on R1-Distill-1.5B as they show usually continued progress even in math problems. Therefore, I would say the experiments are not providing the significant signal that this algorithm scales. In terms of the reasoning to support the conclusion that mitigating the dominance of low-probability tokens is key in RLVR, while the K&K results support the claim, we don't know if this holds in more serious scenarios like math problems. I don't know if K&K is considered a task that its performance forecasts the performance on downstream hard reasoning.
>
> Regarding your concerns on math-related tasks, we do agree with your point that our methods achiveves only slightly better results in comparison with the baseline GRPO. This may attribute to the fact that such math-related dataset may be mixed into the pre-training dataset of the base model, when doing RL, it converges very fast within 100 steps thereby the difference is hard to tell.
>
> Thank you for the insightful suggestion. Following your recommendation, we provide additional experimental results with DeepSeek-R1-Distill-1.5B on the ORZ-math dataset in Table 10 and Figure 18 on Page 29 of the updated version of the paper. These results indeed demonstrate a continuous learning trend throughout the training process. Through numerical analysis in the table, we find that although the performance gains of our proposed methods are not as substantial as those on the K&K dataset, the improvements remain clear and considerable. This is more evident in the figures, where our proposed methods surpass the GRPO baseline throughout almost the entire training process, in both the reward curves and the test-score curves.
>
> Furthermore, to demonstrate that our methods generalize to different models and different policy-gradient-based RL algorithms, we provide additional experiments on LLaMA-series models (Table 9, Figures 16-17, Page 27-28) and introduce a new baseline algorithm, DAPO (Table 8, Figures 14-15, Page 25-27). **For a quick overview of the experimental results, please refer to our global response for more details.**
>
> ### Questions
>
> 1. Is there evidence that K&K is a hard reasoning benchmark?
>
> First, for math-related datasets, many current open-source LLMs have been exposed to highly similar data during their pre-training phase, which leads to extremely rapid convergence of RL training on these datasets. In contrast, for the K&K Logic Puzzle dataset, such relatively niche data is highly unlikely to have been encountered during the models' pre-training phase. Consequently, almost all models demonstrate a continuous learning process on this dataset. Therefore, we believe that compared to math datasets, K&K serves as a more equitable benchmark for validating RL algorithms.
>
> Second, as shown in Table 1 (Page 7), even SOTA reasoning models (GPT-4o/o1/DeepSeek-R1) exhibit poor performance on the test set of this dataset, indicating its high level of difficulty. Furthermore, the small models we employ (with fewer than 8B parameters) can surpass these much larger reasoning models after RL training, which better highlights the enhancement effect of RL algorithms on model reasoning performance.
>
> 2. Have you tried this method on native reasoning models, not base models, and observe improvement?
>
> Following your suggestions, we have supplemented the experiments with DeepSeek-R1-Distill-1.5B. Please refer to our response to the weaknesses you listed above, and the global resposne Point 5.

---

### Author Response · Authors · 2025-11-19
**Response to All Reviewers**

We would like to express our sincere gratitude to the reviewers for their dedicated time and effort in reviewing our work. We deeply appreciate the professional and constructive feedback provided and are encouraged by the recognition of the novelty and significance of our contributions. Below, we provide detailed responses and clarifications addressing each reviewer's suggestions and comments.

We have also revised our manuscript to address all reviewers' concerns. All modifications in the revised manuscript are highlighted in blue. A summary of the key revisions is provided below:

1. We added a new paragraph in Appendix B (Page 17) to provide detailed guidance on the selection of hyperparameters related to our proposed methods ($\alpha$ for *Advantage Reweighting* and $\eta$ for *Lopti*).
2. We added a new subsection in Appendix C.3 (Page 22-23) to explain why the combination of AR and Lopti does not yield improved results for mathematics-related tasks.
3. We added a new subsection in Appendix D.1.2 (Page 25-27) to present experimental results using a new baseline algorithm, DAPO [1], on the K&K Logic Puzzles dataset. For quick reference, we summarize the experimental results in the table below.


| Model | Num | of | People | | | Avg. |
|-|-|-|-|-|-|-|
|  | 3 | 4 | 5 | 6 | 7 |  |
| **Qwen2.5-3B-Instruct** | 0.09 | 0.10 | 0.03 | 0.05 | 0.02 | 0.06 |
| + DAPO | 0.68 | 0.61 | 0.45 | 0.42 | 0.31 | **0.49** |
| + DAPO + Reweight | 0.72 | 0.68 | 0.52 | 0.51 | 0.47 | **0.58** (↑17.4%) |
| + DAPO + Lopti | 0.77 | 0.73 | 0.60 | 0.58 | 0.50 | **0.64** (↑24.6%) |
| + DAPO + Reweight + Lopti | 0.85 | 0.88 | 0.71 | 0.70 | 0.58 | **0.74** (↑50.6%) |
| **Qwen2.5-7B-Instruct-1M** | 0.22 | 0.15 | 0.08 | 0.10 | 0.02 | 0.11 |
| + DAPO | 0.96 | 0.93 | 0.81 | 0.76 | 0.67 | **0.82** |
| + DAPO + Reweight | 0.97 | 0.97 | 0.92 | 0.87 | 0.86 | **0.92** (↑11.1%) |
| + DAPO + Lopti | 0.99 | 0.99 | 0.97 | 0.92 | 0.90 | **0.95** (↑15.5%) |
| + DAPO + Reweight + Lopti | 1.00 | 0.99 | 0.98 | 0.96 | 0.91 | **0.97** (↑17.2%) |

4. We added a new subsection in Appendix D.2.1 (Page 27-28) to present experimental results using LLaMA-series models on the K&K Logic Puzzles dataset. A summary of the experimental results is provided in the table below.

| Model | Num | of | People | | | Avg. |
|-|-|-|-|-|-|-|
|  | 3 | 4 | 5 | 6 | 7 |  |
| **LLaMA-3.2-3B-Instruct** | 0.00 | 0.00 | 0.01 | 0.00 | 0.01 | 0.00 |
| GRPO | 0.47 | 0.33 | 0.29 | 0.27 | 0.13 | **0.30** |
| GRPO + Reweight | 0.58 | 0.51 | 0.33 | 0.36 | 0.23 | **0.40** (↑33.3%) |
| GRPO + Lopti | 0.73 | 0.70 | 0.59 | 0.49 | 0.43 | **0.59** (↑96.7%) |
| GRPO + Reweight + Lopti | 0.73 | 0.76 | 0.58 | 0.55 | 0.45 | **0.62** (↑106.7%) |
| **LLaMA-3.1-8B-Instruct** | 0.05 | 0.01 | 0.03 | 0.00 | 0.00 | 0.02 |
| GRPO | 0.86 | 0.88 | 0.77 | 0.72 | 0.68 | **0.78** |
| GRPO + Reweight | 0.89 | 0.92 | 0.86 | 0.81 | 0.78 | **0.85** (↑9.0%) |
| GRPO + Lopti | 0.90 | 0.95 | 0.89 | 0.87 | 0.84 | **0.88** (↑12.8%) |
| GRPO + Reweight + Lopti | 0.94 | 0.97 | 0.89 | 0.86 | 0.82 | **0.90** (↑15.4%) |

5. We added a new subsection in Appendix D.2.2 (Page 28-29) to present experimental results using the DeepSeek-R1-Distill-1.5B model on the ORZ-math dataset. The key results are presented in the table below.

| Dataset | Algorithms | Olympiad Bench | Minerva | MATH 500 | AMC avg@16 | AIME24 pass@16 | AIME24 avg@16 | Avg. all |
|-|-|-|-|-|-|-|-|-|
| **DeepSeek-R1-Distill-1.5B** | | 24.81 | 9.93 | 55.40 | 34.64 | 23.33 | 9.58 | 26.28 |
| **ORZ** | + GRPO | 35.84 | 19.31 | 73.90 | 49.85 | 33.26 | 14.78 | **37.86** |
|  | + GRPO + Reweight | 40.02 | 19.73 | 77.20 | 53.97 | 32.22 | 17.50 | **40.11** |
|  | + GRPO + Lopti | 37.25 | 20.34 | 75.67 | 51.66 | 34.44 | 14.72 | **39.01** |

[1]. Yu et al. DAPO: An Open-Source LLM Reinforcement Learning System at Scale.

---

### Meta-Review · Area_Chair_mBWG · 2026-01-07

**Summary:**

This paper identifies that low-probability tokens give an outsized effect on RL fine-tuning of LLMs, and provides some methods to help avoid these problems. As is common in this general area, substantial improvements are seen on Knights and Knaves puzzles, with much more modest (but extant) improvements on general math datasets.

The most important reviewer complaints, to my mind, have all been addressed. Given the combination of scientific insight and improved practical performance, I am happy to recommend acceptance of this paper.

While the setting is quite different, some of the points made here remind me of [Heavy-Tailed Class Imbalance and Why Adam Outperforms Gradient Descent on Language Models](https://arxiv.org/abs/2402.19449) (NeurIPS-24) – it might be good to mention this in the related work. In addition to your (nice, thorough) discussion of the relationship to the 80/20 rule, also probably relevant to mention are the relationship to [Gradient Imbalance in DPO](https://arxiv.org/abs/2502.20847) (arXiv only) and [On The Effect of Negative Gradient in GRPO](https://openreview.net/forum?id=2K9QsDaqkM) (NeurIPS-25).

**Reviewer Concerns:**

Concerns about the number of models and benchmarks have been addressed in the rebuttal. So have concerns about the computational overhead, with a nice table comparing overhead in training and sampling; two reviewers addressed hyperparameter concerns, which also seem addressed-enough to me here.

**Reviewer Scores:**

Since most issues were addressed to my satisfaction, I assume they also would have been to the reviewers' satisfaction and scores would have increased somewhat.

---

### Decision · Program_Chairs · 2026-01-26

Accept (Poster)